
# A satellite and reanalysis view of cloud organization, thermodynamic, and dynamic variability within the subtropical marine boundary layer

Brian H. Kahn[1], Georgios Matheou[1], Qing Yue[1], Thomas Fauchez[2], Eric J. Fetzer[1], Matthew Lebsock[1], João Martins[3], Mathias M. Schreier[1], Kentaroh Suzuki[4], and João Teixeira[1]

[1] Jet Propulsion Laboratory, California Institute of Technology, Pasadena, CA, USA
[2] NASA Goddard Space Flight Center, Greenbelt, MD, USA
[3] Instituto Português do Mar e da Atmosfera, Lisbon, Portugal
[4] Atmosphere and Ocean Research Institute, The University of Tokyo, Kashiwa, Japan

*Correspondence to*: Brian H. Kahn (brian.h.kahn@jpl.nasa.gov)

**Abstract.** The global-scale patterns and covariances of subtropical marine boundary layer (MBL) cloud fraction and spatial organization with atmospheric thermodynamic and dynamic fields remain poorly understood. We describe a novel approach that leverages coincident NASA A-train and the Modern Era Retrospective-Analysis for Research and Applications (MERRA) data to quantify the relationships in the subtropical MBL derived at the native pixel and grid resolution. Four subtropical oceanic regions that capture transitions from closed-cell stratocumulus to open-cell trade cumulus are investigated. We define stratocumulus and cumulus regimes based exclusively from infrared-based thermodynamic phase. Visible reflectances are normally distributed within stratocumulus and are increasingly skewed away from the coast where disorganized cumulus dominates. Increases in MBL depth, wind speed and effective radius ($r_e$), and reductions in 700-1000 hPa moist static energy differences and 700 an 850 hPa vertical velocity, correspond with increases in reflectance skewness. We posit that a more robust representation of the cloudy MBL is obtained using visible reflectance rather than retrievals of optical thickness that are limited to a smaller subset of cumulus. An increase in $r_e$ within shallow cumulus is strongly related to higher MBL wind speeds that further correspond to increased precipitation occurrence according to CloudSat. Our results are consistent with surface-based observations and suggest that the combination of A-train and MERRA data sets have potential to add global context to our process understanding of the subtropical cumulus-dominated MBL.

## 1 Introduction

Much of the uncertainty in projections of future climate is directly or indirectly related to clouds and their associated processes (IPCC AR5, 2013) including shallow marine cumuliform clouds (Bony and Dufresne, 2005). The low cloud-climate feedback is generally regarded to be positive (e.g., Clement et al., 2009). Many studies however suggest that the sign and magnitude of the feedback are cloud-type dependent (e.g., Caldwell et al. 2013; Bretherton et al., 2013; Brient and Bony, 2013; Dal Gesso et al., 2015; Stephens, 2005; Yue et al., 2016; Zelinka et al., 2012).



Using large eddy simulation (LES) experiments forced with doubled $CO_2$, Bretherton et al. (2013) show that the gradient of RH from the MBL to the free troposphere is a key factor that controls the shortwave cloud radiative feedback. Rieck et al. (2012) used LES forced by perturbed lower tropospheric temperature profiles with fixed RH to show that an increase in surface moisture fluxes leads to a drying of the trade cumulus-topped MBL. The drying overwhelms the increased shortwave reflection from the liquid water lapse rate feedback, thus leading to reduced cloudiness and a positive shortwave cloud feedback. These mechanisms are also discussed by Nuijens and Stevens (2012) in the context of bulk theory and clearly demonstrate that free tropospheric temperature and moisture gradients act as constraints for climate change-induced surface flux changes.

While the constant RH framework is a useful concept to investigate cloud-climate feedback in simplified modeling experiments, an overall reduction of RH in the subtropical free troposphere was found in the CMIP3 (Sherwood et al., 2010; Fasullo and Trenberth, 2012) and CMIP5 (Lau and Kim, 2015) archives with a non-negligible spread in the changing magnitude and vertical structure of RH among the models. Therefore, the assumption that constant RH might hold across the diversity of subtropical cloud regimes with a changing climate is likely not valid. Medeiros and Nuijens (2016) showed that the RH gradient between the MBL and lower free troposphere is widely variable among the CMIP5 models within the trade cumulus regime. Sandu and Stevens (2011) show that the rate at which the transition from stratocumulus to trade cumulus occurs is modulated by downwelling longwave flux that is in part driven by free tropospheric humidity variations, but the overall morphology of the transition is fairly insensitive to the humidity. Christensen et al. (2013), Bretherton et al (2013), and other works confirm the importance of free tropospheric moisture on the MBL through increased downward longwave emission that helps modulate the rate of cloud-top entrainment. Therefore, further examination of cloud organization and the vertical structure of RH with present-day satellite and reanalysis observations is warranted.

A strong linkage between cloud amount and EIS (Wood and Bretherton, 2004), lower tropospheric stability (LTS) (Klein and Hartmann, 1993), and moist static energy differences (dMSE) between the free troposphere and surface (Kawai and Teixeira, 2010; Chung et al., 2012; Kubar et al., 2015) is well understood. Satellite observations of the MBL have revealed prodigious variations of cloud organization that span orders of magnitude over spatial and temporal scales (Cahalan et al., 1994; Atkinson and Zhang, 1996; Wood and Hartmann, 2006; Muhlbauer et al., 2014). Even for a fixed value of cloud fraction, a large diversity of cloud spatial organization may be observed (Kawai and Teixeira, 2012). Correlations of cloud fraction to other environmental variables are highly dependent on the time scale of comparison (e.g., Brueck et al., 2015). At present, the relationships of cloud fraction and spatial organization to larger-scale properties other than EIS/LTS remain poorly understood. Furthermore, previous work has emphasized correlations of MBL cloud properties to 500 hPa vertical velocity and RH that are averaged over monthly, seasonal, or annual time scales. Kawai and Teixeira (2010) found significant correlations for instantaneous observations of cloud inhomogeneity and the skewness of LWP to thermodynamic structure changes between 850 and 1000 hPa; the correlations are larger for LWP than with cloud fraction.

Modeling and observational studies have demonstrated that that the vertical structures of moments of conserved thermodynamic variables depend on the cloud regime (e.g., Suselj et al., 2013; Ghate et al., 2016). Zhu and Zuidema (2009)



used cloud-resolving models forced by field campaign data to quantify the statistical variability of thermodynamic and dynamic properties of the subtropical MBL. There are substantial differences between stratocumulus and trade cumulus in the mean, variance, skewness, and kurtosis of equivalent potential temperature $\theta_e$, liquid water potential temperature $\theta_l$, and vertical velocity profiles. The differences that exist among the modeled cloud regimes point to the importance of obtaining a

global perspective uniquely provided by satellite and reanalysis data. The NASA A-train (Stephens et al., 2002) provides a wealth of remote sensing data about the microphysics and thermodynamics of the cloudy MBL. Model reanalysis data such as the Modern Era Retrospective-Analysis for Research and Applications (MERRA; Rienecker et al., 2011) offer a complementary set of thermodynamic and dynamic variables that help establish a larger-scale perspective for coincident remote sensing observations. Reanalysis data is typically used in the context of gridded monthly, seasonal or annual means

and the native spatial and temporal resolution available has not been well explored at this point.

We describe a method that blends together the reanalysis and remote sensing data sets at the native temporal and spatial resolution of the observations. This approach has been successfully implemented in reconciling multiple satellite cloud products (Nasiri et al., 2011) and improving the robustness of satellite radiance inter-comparisons (Schreier et al., 2010). The matching approach uses a nearest neighbor technique weighted by the sensor spatial response function (Schreier et al.,

2010). The mean, variance, and skewness of MODIS cloud properties at 1-km or 5-km resolution is retained within a larger 45-km resolution Atmospheric Infrared Sounder (AIRS)/Advanced Microwave Sounding Unit (AMSU) field of regard (FOR), while MERRA's $1/2° \times 2/3°$ resolution thermodynamic and dynamic variables are matched to the nearest AIRS/AMSU FOR. The geophysical fields are retained at the native spatial and temporal resolution such that the instantaneous "snapshots" of the cloud probability density function (pdf) are preserved and are then conditioned by available

thermodynamic and dynamic variables. The statistical behavior of cloud properties and their spatial organization, and how the thermodynamic and dynamic state variables are related to them, are inferred using the finest temporal and spatial resolutions available.

Section 2 describes the data sets used while Section 3 details the methodological approach taken in this investigation. Section 4 details the results, beginning with a regional spatial context, then concluding with examination of joint pdfs. We

conclude in Section 5.

## 2 Data

The AIRS/AMSU sounding suite located onboard NASA's EOS Aqua satellite has obtained vertical profiles of temperature and water vapor at approximately 45-km horizontal resolution since September 2002 (Chahine et al., 2006). Kalmus et al. (2015) describe comparisons of AIRS temperature and water vapor profiles to radiosondes launched during the Marine ARM

GPCI of Clouds (MAGIC) campaign and ECMWF analysis data. While AIRS cannot capture the sharpness of the temperature and water vapor mixing ratio gradients across the top of the MBL (Maddy and Barnet, 2008; Yue et al., 2011), the coarse-resolution vertical gradients from the surface to the lower free troposphere are obtained with high fidelity (Yue et





al., 2013; Kalmus et al., 2015). The AIRS operational products also provide numerous cloud variables that include effective cloud fraction (ECF), cloud thermodynamic phase (liquid, ice, and unknown categories), and others (Kahn et al., 2014). A MBL depth estimate inferred from the height/pressure of maximum RH gradient is described and validated with radiosondes launched during the Rain in Shallow Cumulus over the Ocean (RICO) campaign in Martins et al. (2010).

The Version 5 AIRS channel 4 reflectance (0.49–0.94 μm) (Gautier et al., 2003; Aumann et al., 2006) with a nadir spatial resolution of 2.28 km is used. AIRS visible band data is co-registered to the AIRS IR footprint such that 72 visible pixels are aligned within every footprint. A prototype AIRS visible cloud mask (Gautier et al., 2003) that was developed to support earlier algorithm development efforts is also used. Although the cloud mask has not been compared directly against benchmarks such as the MODIS cloud mask, manual inspection suggests that this cloud mask is clear-sky conservative and
captures many shallow, broken sub-pixel cumulus clouds.

The Moderate Resolution Imaging Spectroradiometer (MODIS) instrument on EOS Aqua is capable of observing a wide variety of land, ocean, and atmospheric variables (Platnick et al., 2003, 2017) that are co-located to the AIRS FOV (Schreier et al., 2010). We use the Collection 5.1 liquid phase cloud optical thickness $\tau$ and effective radius $r_e$ retrievals from the MYD06_L2 swath product and the 1-km cloud mask from the MYD035_L2 swath product. Zhang et al. (2012, 2016) and
Cho et al. (2015) have shown that the retrievals behave well within homogeneous clouds but contain biases along cloud edges and within partly cloudy pixels where 3-d radiative effects become important and algorithmic assumptions are challenged. Platnick et al. (2017) show that the re change between C5.1 and C6 is ±1–2μm. We have tested the differences in the pdfs between C5.1 and C6 for a subset of the data investigated and very little change in the pdfs were observed (not shown). The MODIS liquid cloud $r_e$ is used as a proxy for precipitation and is verified with the CloudSat 2C-RAIN-
PROFILE (Release 4) precipitation product (L'Ecuyer and Stephens, 2002).

The MERRA instantaneous, six-hourly, native-resolution, gridded data sets at $1/2° \times 2/3°$ (Rienecker et al., 2011) are used to assess the thermodynamic profiles derived from AIRS, assign vertical profiles of horizontal u and v wind components, and vertical profiles of pressure velocity ω in the MBL and lower free troposphere. All of the instantaneous MERRA data are spatially and temporally matched to the A-train orbit using a nearest neighbour matching approach with no time
interpolation. Previous investigations have compared climatological averages of MERRA against AIRS (Tian et al., 2013) and GPS-RO (Vergados et al., 2015) in the subtropical MBL. Instantaneous, cloud-regime comparisons to AIRS have not been attempted until the present investigation and the value of this approach is demonstrated in Section 4. We use the 1000, 925, 850, and 700 hPa levels to compare to the AIRS Standard Level 2 product.

### 3 Methodology

Four subtropical oceanic regions that capture transitions from organized closed-cell stratocumulus to disorganized open-cell trade cumulus are investigated (Muhlbauer et al., 2014). The four regions are greatly expanded in scale from those used in Klein and Hartmann (1993) to investigate the stratocumulus-topped MBL and are listed in Table 1. While all available





daytime (ascending) orbits from 1 January 2009 to 31 December 2009 were analyzed in all regions, the remaining discussion is limited to the seasons that contain the observed peak in cloud frequency listed in Table 1 (Klein and Hartmann, 1993).

Figure 1a is an example visible image for a six-minute AIRS granule within the southeast Atlantic Ocean (SEA). The visible band captures various spatial scales of organization. The cloud mask derived from AIRS visible bands for the same granule

is shown in Fig. 1b. The cloud mask is used to narrow down the spatial sampling for the following analysis. The cloud mask likely includes instances of clear sky, but the approach only requires a coarse masking approach to filter out a majority of the clear sky pixels. We will discuss implications regarding the filtering process in Section 4.

Removal of pixels containing mid- and high-level clouds helps to reduce ambiguities introduced by free tropospheric clouds and also a portion of the thermodynamic and dynamic variability associated with cloudy areas of synoptic-scale waves.

Figure 2 shows the AIRS infrared $T_b$ within a clean atmospheric window at 1231 cm$^{-1}$, the cloud thermodynamic phase mask, three constant pressure levels of AIRS RH (700, 850, and 925 hPa), and the skewness of visible reflectance for the same granule shown in Fig. 1. The cloud thermodynamic phase identifies some scattered ice in the northern portion of the granule. All pixels identified with ice are removed in the following analysis. Jin and Nasiri (2014) showed that AIRS successfully identifies the presence of ice within the AIRS FOV in excess of 90% of the time when compared to CALIPSO

thermodynamic phase estimates. The rate of agreement depends on the complexity of the vertical structure and horizontal heterogeneity (Jin and Nasiri, 2014). A similar approach is taken in Nam et al. (2012) and Myers and Norris (2015) to minimize impacts from convection and synoptic-scale weather systems. Additional occurrences of $T_{b,1231} < 273$ K that potentially contain supercooled liquid phase mid-level clouds are also removed in the following analysis.

The AIRS liquid detections coincide with uniform stratocumulus (Fig. 1) with close to normally distributed visible

reflectances (lower right, Fig. 2), while unknown detections correspond well to disorganized shallow cumulus (Muhlbauer et al., 2014) with a distinctive positively skewed visible reflectance, very similar to previous results obtained using liquid water path (LWP) (Wood and Hartmann, 2006; Kawai and Teixeira, 2010). Previous investigations have used free tropospheric vertical velocity to separate cloud regime types (e.g., Bony and Dufresne, 2005; Medeiros and Stevens, 2011; Nam et al., 2012). Henceforth, the two regimes defined exclusively by *liquid* and *unknown* phase detections will be generically referred

to as *stratocumulus* and *cumulus* regimes, respectively. An advantage of this approach is that the temporal and spatial variations of cumulus and stratocumulus cloud areas are more precisely separated from each other.

For the AIRS/AMSU FORs containing MBL clouds, we collocate the coincident AIRS and MODIS geophysical fields. The AIRS ECF is averaged over the entire AIRS/AMSU FOR where clear sky is equal to a value of zero. The AIRS thermodynamic phase is averaged over cloudy AIRS FOVs only. The individual phase tests are summed and we define

liquid for values < -0.8, unknown between -0.8 to +0.8, and ice for values > +0.8. The MODIS cloud mask and $\tau$ are averaged over the entire AIRS/AMSU FOR. The MODIS $r_e$ is averaged only over the successful retrievals that are a subset of MODIS pixels identified as containing cloud. The nearest neighbour is matched for MERRA geophysical fields at a similarly sized (or larger) spatial resolution. The mean, standard deviation, and skewness of MODIS and AIRS FOV cloud properties, visible reflectance, and infrared radiance are then calculated for each AIRS/AMSU FOR separately. Therefore,



multiple satellite instrument and reanalysis observations at multiple spatial scales can be linked together through joint pdfs for a large combination of statistical moments. These data serve as the basis of the following investigation.

## 4 Results

In Section 4.1, regional-scale, seasonal averages are calculated from the pixel-scale data described in Section 3 for 90

daytime (130 pm equatorial crossing time) snapshots and are then re-gridded to 1°×1° spatial resolution. In Section 4.2, multivariate pdfs are investigated in the context of limiting the plethora of choices among variables and statistical moments. In Sections 4.3-4.6, several sets of thermodynamic and microphysical pdfs are quantified and described.

### 4.1 Regional spatial averages

Figure 3 shows the reflectance skewness for JJA in the NEP and NEA regions, and SON in the SEP and SEA regions, with

an overlay of AIRS total ECF. The coastal stratocumulus reflectances are distributed approximately normally while the reflectances are positively skewed away from the coast where disorganized cumulus dominates (e.g., Wood and Hartmann, 2006). Contours of the magnitude of reflectance skewness closely align to the magnitude of ECF in cumulus while much less so in proximity to the coast within stratocumulus. Very poor spatial correspondence between reflectance skewness and the mean value of MODIS cloud fraction was found (not shown) and is consistent with low correlations between GOES derived

cloud fraction and LWP noted by Kawai and Teixeira (2010) in the SEP region.  Interestingly, the average reflectance skewness is larger and ECF is smaller in the NEP than the other three regions and is consistent with other satellite observations (Klein and Hartmann, 1993; Rossow and Schiffer, 1999) and surface-based observations (Wood, 2012). The patterns of reflectance skewness shown in Fig. 3 also resemble typical climatological patterns of cloud sizes reported in Wood and Field (2011) and cloud texture as viewed from the Multi-angle Imaging SpectroRadiometer (MISR) (Zhao et al.,

20   2016).

Values of MODIS total water path (TWP) skewness do not show a clear transition from normally distributed to positively skewed values (Weber et al., 2011). This further motivates the removal of mid- and high-level cloud occurrences using the AIRS phase mask that comprise anywhere from 4 to 18% of the total number of FOVs depending on the region of study (Table 1). Oreopoulos and Cahalan (2005) show that the inhomogeneity parameter calculated from MODIS LWP, rather

than TWP, is most homogeneous near the coast and indicates increasing heterogeneity that extends into the cumulus regimes.  We argue that the results of Oreopoulos and Cahalan (2005) are more definitive than those shown in Weber et al. (2011) and more closely resemble the gradients and magnitudes contained within Fig. 3.

The mean MBL depth (Fig. 4) reaffirms a characteristic transition from shallow MBLs (920–970 hPa) near the coast to deeper MBLs (830–880 hPa) to the west and is a well-observed feature of the stratocumulus to cumulus transition (Karlsson

et al., 2010; Teixeira et al., 2011). Closest to the coast, the MBL is shallowest in the NEA and slightly deeper in the NEP. The SEA and SEP are deeper than their NH counterparts with SEP the deepest. The SEP MBL depths agree with VOCALS-



REx in situ radiosonde-derived temperature inversion base heights (Bretherton et al., 2010). Furthermore, the inter-regional differences in MBL depth show consistency with global positioning system-radio occultation (GPS-RO) data (Chan and Wood, 2013).

Differences in the moist static energy (dMSE) between 700 and 1000 hPa are calculated following the approach outlined by

Kubar et al. (2012) and are shown in Fig. 4. The dMSE is calculated from quality-controlled AIRS soundings (PGood ≥ 1000 hPa) and is nearly identical to estimates from ERA-Interim (Kubar et al., 2012). The magnitude of dMSE is larger and positive near the coast in the SH compared to the NH and is somewhat reduced in the NEA region. Yue et al. (2011) showed that values of EIS and LTS obtained from AIRS soundings are lower in the NEA compared to the other three regions and are also consistent with Fig. 4.

Seasonal averages of AIRS $RH_{700}$ with an overlay of the corresponding MERRA-AIRS $RH_{700}$ differences are shown in Fig. 5. Wind vectors depict the mean horizontal flow. Overall, $RH_{700}$ in the SH is lower than the NH while the NEA is the moistest of the four regions and SEP the driest. MERRA is on average moister than AIRS by ~5% in the NH, nearly identical to AIRS in the SEA, and a much more spatially heterogeneous difference is observed in the SEP from the coastal proximity westward between 8–12°S. Tian et al. (2013) showed that at 700 hPa, MERRA is typically biased wet compared

to AIRS in the NEP and NEA and a more complicated spatial pattern in the SEP that is consistent with Fig. 5.

Bretherton et al. (2010) demonstrate that the free troposphere in the SEP westward of 75°W is characteristically very dry (0.1 g kg$^{-1}$) with sporadic filaments of moist air (as high as 3-6 g kg$^{-1}$) up to an altitude of 2.5 km. In addition, these moist filaments have been observed with GPS-RO refractivity profiles (von Engeln et al., 2007). The vertical structure of RH obtained from VOCALS-REx radiosondes implies a well-mixed MBL near the coast with MBL decoupling west of 80°W.

Myers and Norris (2015) showed that 700 hPa is drier in the SH subtropics compared to the NH using ERA-Interim data. When GCMs are sampled for RICO-like conditions using representative mid-tropospheric large-scale vertical velocities (Medeiros and Stevens, 2011), a dry bias is obtained above the MBL in comparison to a composite of RICO radiosondes.

The seasonal averages of AIRS $RH_{850}$, and the corresponding MERRA-AIRS $RH_{850}$ differences, are larger than that found for $RH_{700}$ (Fig. 6) and are due to temperature and water vapor weighting function widths on the order of 2–3 km (Maddy and

Barnet, 2008). While the MBL is typically deepest in the SEP, the magnitude of $RH_{850}$ is lower compared to the NEP and is further evidence for a drier and warmer lower free troposphere in the SEP. The MERRA-AIRS differences are -5% to -10% in the NEA, while they are mostly positive and up to +15% in the NEP, SEP, and SEA.

In summary, the seasonal averages exhibit realistic three-dimensional spatial morphologies and gradients and show consistency with MERRA RH in the subtropical MBL. The MBL depth and seasonal variations (not shown) agree with GPS-

RO (Chan and Wood, 2013). The AIRS-derived dMSE between 700 and 1000 hPa agrees with ERA-Interim (Kubar et al., 2012). The reflectance skewness is strongly related to cloud organization and dMSE (Kawai and Teixeira, 2012). The AIRS ECF distributions closely correspond to well-established climatologies of cloud amount (e.g., Klein and Hartmann, 1993;



Rossow and Schiffer, 1999; Wood, 2012). The vertical structure of the horizontal wind flow well represents known climatological patterns in the MBL and lower free troposphere.

In the following sub-sections, an ensemble of multivariate and multi-moment pdfs are examined, and novel insights on the structure of the cloud-topped subtropical MBL will be discussed.

**4.2 Dimensionality of multivariate pdfs**

Choosing an ideal subset of variables and statistical moments to form the basis of joint histograms is a great challenge. Motivated in large part to link cloud and thermodynamic properties derived from infrared and visible bands, we describe four variable combinations: (1) visible reflectance versus ECF; (2) $\tau$ versus ECF; (3) visible reflectance versus cloud fraction; and (4) $\tau$ versus cloud fraction. The natural log frequency of occurrence for the four combinations is shown in gray
scale from black to white and MBL depth is superimposed as colored and labelled contours (Fig. 7).

The MBL depth exhibits clearer patterns in the ECF dimension rather than the cloud fraction dimension. The latter is more compressed and the gradients are weaker in both dimensions. The MBL depth is deepest for lower values of ECF, $\tau$, and reflectance. In addition, the MBL depth also decreases for the most reflective clouds at a given value of ECF while this behavior is not observed for $\tau$. We posit that an additional population of sub-pixel cumulus clouds is captured within the
reflectance data that is not captured in $\tau$ data.

We will use reflectance versus ECF in the remainder of this work. We are not advocating that the dimensional choices made are optimal. Instead, the results motivate a fresh look at available satellite and reanalysis data, their joint distributions and statistical moments, building from native resolution, pixel-scale, temporally instantaneous coincidences.

**4.3 Regional differences in MBL depth and dMSE**

The occurrence frequencies of AMSU FORs that contain stratocumulus and cumulus are listed in Table 1. The largest differences in the gradients between stratocumulus and cumulus are found in the NEP (Fig. 8a,e), while the smallest differences are found in the NEA (Fig. 8c,g). The MBL depth is several 10s of hPa shallower in stratocumulus (Fig. 8a-d) compared to cumulus (Fig. 8e-h) in all four regions. For almost every possible combination of reflectance and ECF, it is encouraging that the MBL depth is shallower for stratocumulus than cumulus. We can conclude that the cloud amount and
shortwave reflected radiation act independently of MBL depth. A small population of shallow MBL depths for ECF > 0.9 is found in cumulus (Fig. 8e-h) and is a consequence of a small sample size of stratocumulus clouds that fail to exhibit a large enough $T_b$ signature to trigger liquid phase tests (e.g., Kahn et al., 2011; 2014). The two cloud regimes therefore should not be considered mutually exclusive of each other.

A significant increase in MBL depth with increasing reflectance is found in cumulus with a stronger relationship noted in the
NH compared to the SH (Fig. 8e-h) at a fixed value of ECF. This is partly a result of a deeper MBL in the SEP and SEA near the coastline (Fig. 4). The exception is that the NEP, SEP, and SEA all show a decrease for the most reflective clouds except



for the NEA. The MBL depth gradients demonstrate a more linear relationship with the standard deviation of reflectance (Fig. 8i-l) than the average reflectance (Fig. 8e-h). The MBL is deepest for the largest values of the standard deviation at almost all values of ECF in all four regions. This suggests that the largest values of average reflectance in Fig. 8e-h are uniform in spatial structure and have among the lowest standard deviations (Fig. 8e-h).

The reflectance skewness is shown in Fig. 8m-p. There are several important features to describe. First, the MBL depth is shallower for normally distributed reflectance and a sharp increase in MBL depth with increasing positive skewness is consistent with Figs. 3 and 4. Second, the change in MBL depth is somewhat greater for an identical increase in reflectance skewness when compared to $\tau$ skewness (not shown). Third, the population of cumulus occurrences at low ECF for positive skewness > 1 are mostly absent in the $\tau$ data (not shown) but are very common in reflectance data. We argue that this

discrepancy has an important impact on the interpretation of the trade cumulus climatology. The gradient of MBL depth in the dimension of increasing positive skewness at low values of ECF is much greater in the reflectance data where the highest data counts are found. We posit that the reflectance data contain more disorganized subpixel cumulus than the $\tau$ data. Fourth (not shown), we remove the AIRS cloud mask filter (Fig. 1b) in order to retain all values of reflectance (clear and cloudy) in the joint pdf. While the patterns of reflectance skewness and MBL depth are not significantly altered when applying the

cloud mask filter, many more counts with normally distributed reflectances appear that indicates some leakage of weak clear-sky surface reflection. We conclude that there is a much bigger difference between the cloud mask-filtered reflectance and $\tau$, rather than between the filtered and non-filtered variants of reflectance, implying a robust interpretation. Fifth, the MBL depth contours change more rapidly with skewness of $\tau$ or reflectance rather than with the mean value of $\tau$ or reflectance, consistent with the findings of Kawai and Teixeira (2010) where a tighter correlation with LWP skewness

compared to average LWP was found.

Figure 9 shows that the dMSE in the SEP is positive in sign and largest in magnitude for larger values of ECF and normally distributed reflectance (other regimes are similar and are not shown). In the case of reflectance skewness, contours of constant dMSE track closely the occurrence frequency through much of the joint pdfs, with a reduction of dMSE to values less than zero at a fixed value of ECF as positively skewed reflectance increases. This behavior is similar to MBL depth (Fig.

8f) and suggests that dMSE is correlated with small-scale spatial organization of clouds rather than the conventional wisdom that it best correlates with larger-scale atmospheric thermodynamic structure. Kawai and Teixeira (2012) showed that the skewness of LWP varies from +1 to +2 for cloud amounts of 90–100%, and up to +1.5 to +3.5 for cloud amounts < 30%. Furthermore, Kawai and Teixeira (2010) found that the highest correlations occur between LWP homogeneity, skewness and kurtosis to different measures of temperature and moisture differences from the surface to 850 hPa, rather than to values of

EIS and LTS.



### 4.4 Vertical structure of RH

Figure 10 shows the three moments of AIRS RH in the cumulus-dominated SEP at four separate levels (700, 850, 925 and 1000 hPa). The driest air is observed at $RH_{700}$, while the moistest is observed at $RH_{925}$ and $RH_{1000}$. Very similar magnitudes and gradients are observed for MERRA RH at all four levels and three statistical moments (Fig. A1). The largest magnitudes

of RH correspond to the deepest MBLs (Fig. 8) and MERRA $RH_{925}$ is as large as 95% (Fig. A1).

Despite known sensitivity limitations of infrared sounding in opaque clouds, and vertical smoothing due to the nature of satellite infrared weighting functions (see Appendix A), there exists a strong consistency between AIRS and MERRA RH in the cloudy subtropical MBL and lower free troposphere. This comparison demonstrates the value of both AIRS and MERRA RH and should lend confidence to the use of both data sets.

### 4.5 Variations of $r_e$

Figure 11 shows the MODIS derived $r_e$ for stratocumulus (Fig. 11a-d) and cumulus (Fig. 11e-p). There are several prominent features in the histograms. First, the stratocumulus $r_e$ is about 11 to 12 μm throughout most of the pdf in all four regions. An exception is the increase of $r_e$ by several μm when average reflectance and ECF are reduced (Fig. 11a-d). While these particular MODIS pixels were successful, cloud horizontal inhomogeneity may cause larger $r_e$ within this population of

clouds because of the plane parallel homogeneous bias (Cho et al., 2015; Zhang et al., 2016). Second, the NEA region (Fig. 11g) is most dissimilar to the other three regions for average (Figs. 11e-h), standard deviation (Fig. 11i-l) and skewness (Fig. 11m-p). Third, the $r_e$ is largest along the axis of maximum counts with values upwards of 16 to 20 μm in the SEP, 15-18 μm in the SEA, and 14–17 μm in the NEP. The largest values in the NEA are confined to the most skewed reflectances unlike the other three regions. Fourth, in the cleaner SH, the values of $r_e$ appear to be more tightly coupled to cloud microphysical

processes that respond to changing wind speed and a deepening MBL (see Section 4.6).

One general interpretation of the larger $r_e$ in cumulus when contrasted to stratocumulus relates to the increased inhomogeneity of cumulus (Zhang et al., 2012, 2016), retrieval failures (Cho et al., 2015), and view angle biases (e.g., Liang et al., 2015) that are coupled together in numerous and complex ways (Cho et al., 2015). As these particular MODIS pixels are limited to successful retrievals only, we offer evidence that the increase in $r_e$ is entirely consistent with environmental

variability that is furthermore consistent with droplet growth and precipitation. The contours of $r_e$ correspond very closely to the magnitude of the u-component of wind speed at 925 hPa ($u_{925}$) (see Section 4.6) and other levels in the MBL (not shown), suggesting a link between cloud droplet growth, light rain, and dynamical variability. The somewhat larger $r_e$ in the SH is consistent with droplet growth in a cleaner environment (Suzuki et al., 2010a,b). Successful retrievals with pixels that have increased subpixel horizontal inhomogeneity could produce larger $r_e$ (Cho et al., 2015), and furthermore could be

correlated with increased precipitation frequency. Zhang et al. (2016) show primarily large $r_e$ in scenes with inhomogeneity, although a smaller fraction of clouds have smaller $r_e$.





To determine if the elevated $r_e$ along the axis of maximum counts is associated with increased precipitation, we use collocated matchups of the CloudSat precipitation rate to determine which AMSU FOVs contain occurrences of precipitation. Figure 12 shows results for the SEP region. The reflectance skewness for the full AIRS/AMSU/MODIS swath in Fig. 11n is restricted to the CloudSat ground track in Fig. 12a. The counts are reduced by a factor in excess of 30 as

expected. There are some subtle changes in the $r_e$ distribution showing an increase of 2-3 μm with increasing skewness at a fixed value of ECF. Figure 12b shows the proportion of the pdf that contains at a minimum the natural log(2) counts of precipitation occurrence within each bin. About 20-50% of the AMSU FOVs are precipitating according to CloudSat within the pdf of Fig. 12a. The precipitation frequency is consistent with Rapp et al. (2013) where up to 40% of clouds precipitate in the cumulus regime. Little to no precipitation occurs outside of the central portion of the pdf in Fig. 12a. The highly

skewed cumulus with ECF<0.2 appear to be exhibiting large $r_e$ biases due to reflectance inhomogeneity (Cho et al., 2015; Zhang et al., 2016). We also point out that the population of clouds detected by CloudSat that have ECF>0.95 (Fig. 12b) are associated with very little precipitation and is consistent with the spatial distributions described by Rapp et al. (2013).

### 4.6 Variations of u and ω

Nuijens et al. (2009) describe Rain in Cumulus over the Ocean (RICO) field campaign observations that illustrate

fundamental physical relationships between cloud cover, wind speed and direction, the vertical structure of RH, and precipitation frequency and intensity within precipitating shallow trade cumulus. The observations can be grouped into three fairly distinct cumulus regimes: (i) low cloud fraction with little to no precipitation characterized by low values of u and a drier MBL; (ii) an increase in cloud fraction with some light precipitation characterized by low values of u and elevated RH between 800-1000 hPa; (iii) a further increase in cloud fraction with light precipitation and some isolated heavier events

characterized by higher values of u and a large increase in RH between 650-900 hPa. A key observational difference among the three regimes is the variation of RH within the MBL (800-1000 hPa), and near the top of the MBL extending into the lower free troposphere (650-900 hPa). The width of these layers is similar to the AIRS 700 and 925 hPa temperature and specific humidity weighting functions. Even though the RICO observations do not fall within any of the four regions listed in Table 1, Medeiros and Nuijens (2016) show that the observational site is applicable to the trade regime as a whole across the

globe. Thus our approach is to determine if similar relationships shown in Nuijens et al. (2009) exist in cumulus for the regions listed in Table 1.

Figure 13 shows $\theta_{e,700}$, $u_{925}$, $\omega_{700}$, and $\omega_{925}$. The $\theta_{700}$ (not shown) is nearly identical among all regions with $\theta_{700}$=314 K ± 1 K. Thus, the structure in $\theta_{e,700}$ (Fig. 13a-d) and $RH_{700}$ is driven by variations in specific humidity. For a fixed value of ECF, the clouds with the lowest and highest values of reflectance are associated with moistening of the lower free troposphere. Using

climatological averages, Myers and Norris (2015) show that shortwave observations from CERES, cloud fraction estimates from ISCCP and CALIPSO, and $RH_{700}$ and $\omega_{700}$ from ERA-Interim reflect aspects of Fig. 13 and, namely, that more reflected shortwave is associated with increased cloud fraction and decreased $\omega_{700}$.





The highest values of $\theta_{e,925}$ (not shown) occur along the axis of highest counts while reductions in $\theta_{e,925}$ occur for the least and most reflective clouds at a fixed value of ECF. This is the case for the NEP, SEP, and SEA, but the NEA is an outlier and shows a constant increase as seen with RH and MBL depth. Unlike 700 hPa, $\theta_{925}$ is more variable (not shown) between the four regions but is generally 2 K or less.

The $u_{925}$ is largest (Fig. 13e-h) when the pdf has the largest counts and very closely resembles $r_e$ in Fig. 11e-h. The subtle differences in the contours in Fig. 11e-h and Fig. 13e-h align very well, suggesting a tight correlation between the two parameters. The magnitude of $u_{925}$ is larger than $u_{700}$ (not shown) consistent with RICO (Nuijens and Stevens, 2009). The $\omega_{700}$ fields (Fig. 13i-l) exhibit minimal correspondence with average reflectance and ECF in the NH regimes with a weak correspondence in the average reflectance in the SH regions. The $\omega_{925}$ fields (Fig. 13m-p) show larger gradients in all four

regions. The $\omega_{925}$ decreases with increasing reflectance in all regions similar to that shown in Myers and Norris (2015), with a slightly noisier pattern in $\omega_{925}$ observed in the NH regimes. The decrease of $\omega_{925}$ with increasing reflectance is consistent with a deeper MBL (Fig. 8e-h) and larger $\tau$. Where $u_{925}$ (Fig. 13e-h) increases, $\omega_{925}$ (Fig. 13m-p) decreases and RH$_{925}$ increases (Fig. 10c for the SEP only). The largest values of $r_e$ (Fig. 11e-h) also correspond to the above tendencies, consistent with the concept of more frequent precipitating clouds within a windier and deeper MBL (Nuijens and Stevens,

15  2012).

The joint pdfs imply simultaneous increases in $\theta_{e,700}$, $\theta_{e,925}$ (and by extension RH$_{700}$ and RH$_{925}$), $u_{925}$, and ECF in three of the four regions investigated (NEP, SEP, and SEA) with a particularly strong relationship between $u_{925}$ and $r_e$. The NEA is somewhat of an outlier although this is based on one seasons' worth of data during 2009. There is much variability across the trade cumulus regime as it should not be treated as a single homogeneous entity. The satellite and reanalysis observations are

able to quantify aspects of the non-uniqueness between free tropospheric and MBL humidity, cloud coverage, wind speed, subsidence, and precipitation frequency.

## 5 Summary and Conclusions

The global-scale relationships of cloud fraction and spatial organization to thermodynamic and dynamic properties of the atmosphere remain poorly understood. The NASA A-train (Stephens et al., 2002) provides a wealth of remote sensing data

about the microphysics and thermodynamics of the cloudy MBL. The Modern Era Retrospective-Analysis for Research and Applications (MERRA; Rienecker et al., 2011) offers a complementary set of thermodynamic and dynamic variables that helps establish context for coincident remote sensing observations. The synergy between satellite and reanalysis data at the native spatial and temporal resolutions available has not been fully exploited to date. We describe a novel approach that leverages coincident reanalysis and remote sensing data at the native resolution of the observations. The spatial organization

of clouds, and the relationship to thermodynamic and dynamic state variables, is thus inferred using the finest temporal and spatial resolutions available.



Four subtropical oceanic regions that capture transitions from organized closed-cell stratocumulus to disorganized open-cell trade cumulus are investigated (Klein and Hartmann, 1993; Muhlbauer et al., 2014). We define two regimes based exclusively on *liquid* and *unknown* cloud thermodynamic phase detections with the AIRS instrument, and generically refer to them as *stratocumulus* and *cumulus* regimes, respectively. The mean, standard deviation, and skewness of MODIS and AIRS
FOV cloud properties and visible reflectances are calculated for each AIRS and MERRA temperature and humidity observation.

As with previous findings, coastal stratocumulus reflectances are approximately normally distributed while the reflectances are positively skewed away from the coast where disorganized cumulus dominates. The reflectance skewness closely aligns to the magnitude of AIRS effective cloud fraction (ECF) in cumulus with less correspondence in stratocumulus. Strong
(poor) spatial correspondence between reflectance skewness and AIRS ECF (MODIS cloud fraction) was found suggesting infrared-based ECF is an unappreciated and potentially valuable diagnostic for MBL cloud characterization. The mean MBL depth derived from AIRS (Martins et al., 2010) shows a characteristic transition from shallow MBLs (920–970 hPa) near the coast to deeper MBLs (830–880 hPa) away from the coast and is a well-observed feature of the stratocumulus to cumulus transition (e.g., Teixeira et al., 2011). RH at 700 hPa in the SH is lower than the NH while the NEA is the moistest of the
four regions and SEP the driest. The AIRS-derived moist static energy differences (dMSE) between 700 and 1000 hPa agree very well with ERA-Interim (Kubar et al., 2012). We find that the reflectance skewness is strongly related to cloud organization and the magnitude of dMSE as previously found by Kawai and Teixeira (2012). For almost every possible combination of reflectance and ECF, it is encouraging that the MBL depth is shallower for stratocumulus than cumulus.

The change in MBL depth is somewhat greater for an identical increase in reflectance skewness when compared to $\tau$
skewness. The population of cumulus occurrences at low ECF for positive skewness > 1 are mostly absent in the $\tau$ data but are very common in reflectance data. This highlights the importance of understanding the sampling from derived Level 2 products compared to Level 1 radiances and reflectances that may capture a fuller range of the geophysical state in different cloud regimes.

The lowest RH is observed at 700 and 850 hPa above stratocumulus, while the highest is observed at 925 hPa in proximity to
shallow cumulus. The difference between MERRA and AIRS RH at 925 hPa increases as reflectance or $\tau$ increases and reaffirms the well-known sampling bias of satellite infrared sounding within the cloudy MBL (Fetzer et al., 2006). While AIRS is unable to sample the moistest filaments within clouds that are opaque to infrared radiation, AIRS is able to replicate relative changes in RH exhibited by MERRA. The strong consistency between AIRS and MERRA RH in the cloudy subtropical MBL and lower free troposphere demonstrates the value of both AIRS and MERRA RH as reference data sets.

The $r_e$ in stratocumulus is about 11 to 12 μm for most values of reflectance and ECF in all four regions of study. For cumulus, $r_e$ ranges anywhere from 12 to 20 μm, with larger $r_e$ for increasing positive skewness especially when ECF is small. The values of $r_e$ are more tightly coupled to cloud microphysical processes that respond to changing MBL wind speed and a deepening MBL. We argue that for these successful MODIS retrievals, the increase in $r_e$ is consistent with increased droplet





growth and hence precipitation occurrence. In the SEP region, we confirm that the elevated values of $r_e$ that correspond with the increased $u_{925}$ are in fact precipitating much more frequently according to CloudSat. Clouds are non-precipitating if they are highly skewed with low values of ECF. This result is consistent with the idea that 3-D radiative effects, cloud inhomogeneity and algorithm assumptions break down in highly inhomogeneous shallow cumulus (Cho et al., 2015; Zhang

et al., 2016).

The Rain in Cumulus over the Ocean (RICO) observations provide an important multi-parameter testing benchmark (Nuijens et al., 2009).  These results are generalized into three types of shallow precipitating cumulus regimes observed during RICO. The joint pdfs imply simultaneous increases in $\theta_{e,700}$, $\theta_{e,925}$, $u_{925}$, and ECF in three of the four regions investigated (NEP, SEP, and SEA) with a strong correspondence between $u_{925}$ and $r_e$. The NEA less clearly follows these behaviors and is an

outlier, although this is based on one seasons' worth of data during 2009. The variability among the different regions of study emphasizes the non-uniqueness among changes in free tropospheric humidity, cloud coverage, wind speed, and subsidence. Our results are consistent with Nuijens and Stevens (2009) and may offer additional insight into the global context of the structure of the subtropical cumulus-dominated MBL when a larger observational record is examined.

### Appendix A

The differences between MERRA and AIRS $RH_{925}$ are greater as reflectance increases and helps quantify the magnitude of the well-known clear-sky sampling bias of atmospheric infrared sounding (Fetzer et al., 2006) within the cloudy MBL. These results reaffirm the notion that AIRS is unable to sample the moistest filaments of the MBL that reside within clouds that are opaque to infrared radiation. Given this tendency, however, AIRS is still able to replicate the changes in RH exhibited by MERRA with respect to the magnitude of reflectance. Furthermore, when the clouds are generally broken or semi-

transparent, the magnitude of AIRS and MERRA $RH_{925}$ is more similar. Lastly, AIRS $RH_{1000}$ is about 75-80% (Fig. 10) and about 80-85% for MERRA (Fig. A1) and are both near typical values for the low latitude oceans (Richter and Xie, 2008). Uniform values of $RH_{1000}$ are found for all combinations of reflectance and ECF.

### Acknowledgments

Part of this research was carried out at the Jet Propulsion Laboratory (JPL), California Institute of Technology, under a

contract with the National Aeronautics and Space Administration. GM and BHK were partially supported by an R&TD project at JPL. BHK was partially supported by the AIRS project at JPL and by the NASA Science of Terra and Aqua program under grant NNN13D455T. BHK, QY, and MMS were partially supported by NASA's Making Earth Science Data Records for Use in Research Environments (MEaSUREs) program. The AIRS version 6 data sets were processed by and obtained from the Goddard Earth Services Data and Information Services Center (http://daac.gsfc.nasa.gov/). The MODIS

collection 6 data sets were processed by and obtained from the Level 1 and Atmosphere and Archive Distribution System





(http://ladsweb.nascom.nasa.gov). The MERRA data sets were processed by and obtained from the NASA Goddard's Global Modeling and Assimilation Office (GMAO). CloudSat data were obtained through the CloudSat Data Processing Center (http://www.cloudsat.cira.colostate.edu/). The data and code used in this investigation is available upon request from the lead author © 2017. All rights reserved. Government sponsorship acknowledged.

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

| Region | Abbrev | Season | Location | % Sc | % Cu | % Other | % Clr |
|---|---|---|---|---|---|---|---|
| Northeast Pacific Ocean | NEP | JJA | 15°N–35°N 110°W–150°W | 23.3 | 64.8 | 10.7 | 1.3 |
| Northeast Atlantic Ocean | NEA | JJA | 15°N–35°N 10°W–50°W | 8.1 | 72.0 | 18.0 | 1.9 |
| Southeast Pacific Ocean | SEP | SON | 5°S–25°S 70°W–110°W | 25.5 | 69.6 | 3.9 | 1.0 |
| Southeast Atlantic Ocean | SEA | SON | 5°S–25°S 25°W–15°E | 31.8 | 62.1 | 4.4 | 1.7 |

**Table 1: The four regions investigated in this study are greatly expanded in area from Klein and Hartmann (1993). The four right**
15 **columns are defined at the AMSU field of regard (FOR) spatial scale. The three cloudy categories indicate whether clouds of that type occur with any frequency within the AMSU FOR. Clear is defined over the entire AMSU FOR and are therefore very infrequent.**



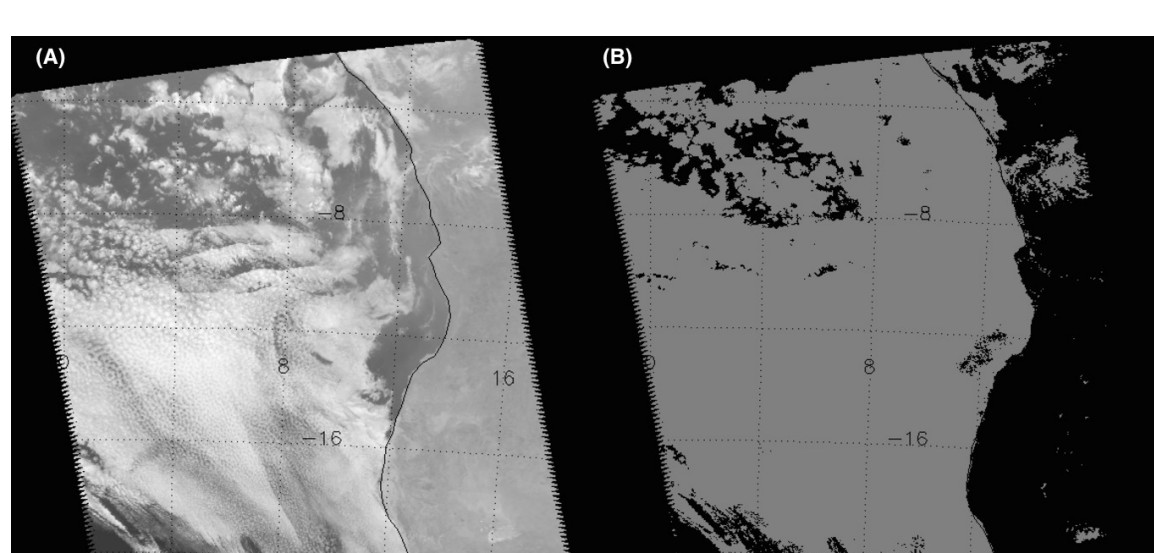

**Figure 1. AIRS version 5 visible channel 4 reflectance (0.49–0.94 µm) at a nadir spatial resolution of 2.28 km (left), and AIRS cloud mask (binary clear and cloudy) determined from visible channel thresholds (right). See** *Gautier et al.* **[2003] for more details.**





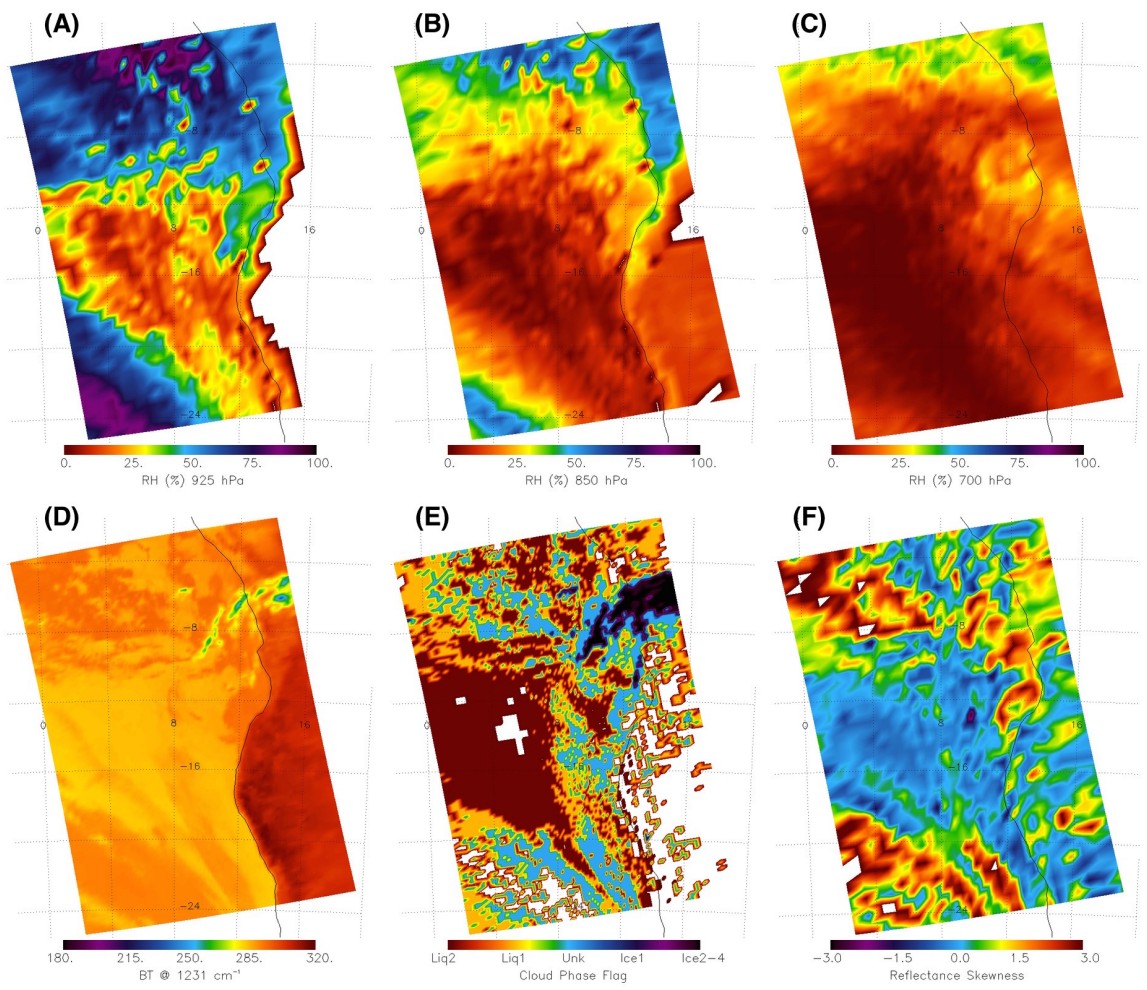

**Figure 2.** AIRS (a) $RH_{925}$ (%), (b) $RH_{850}$ (%), (c) $RH_{700}$ (%), (d) 1231 $cm^{-1}$ $T_b$ (K), (e) cloud thermodynamic phase, and (f) reflectance skewness from visible channel 4. The granule is identical to the one shown in Fig. 1.





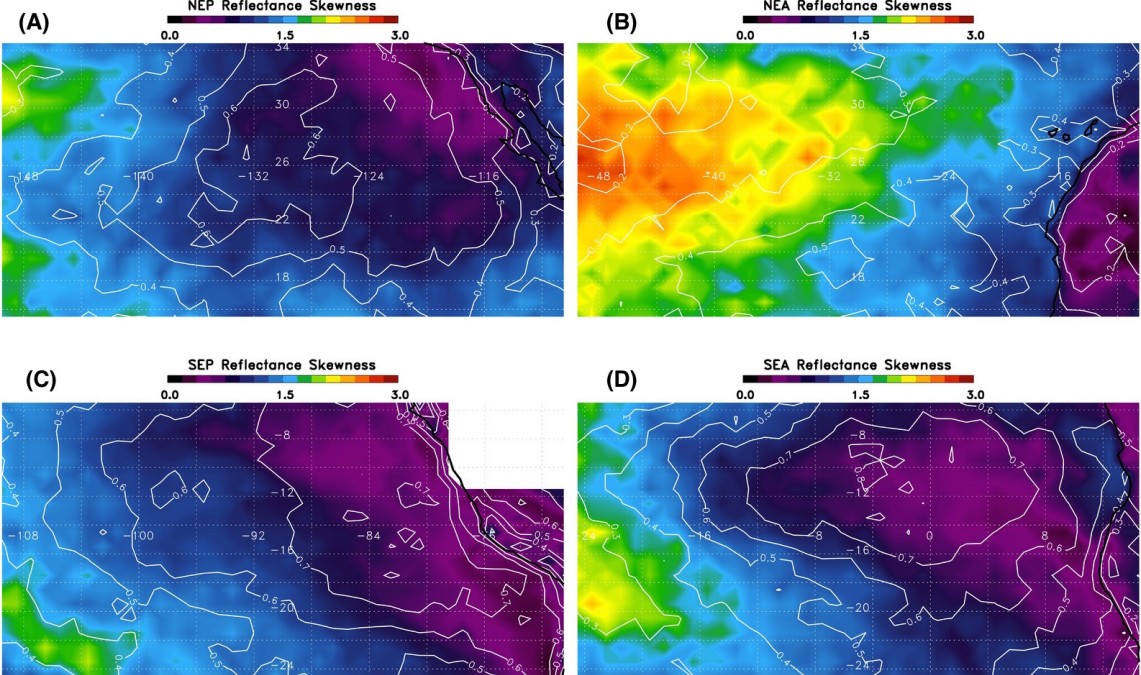

**Figure 3. Reflectance skewness for regions listed in Table 1: (a) NEP, (b) NEA, (c), SEP, and (d) SEA. The AIRS ECF is overlaid as white contours.**





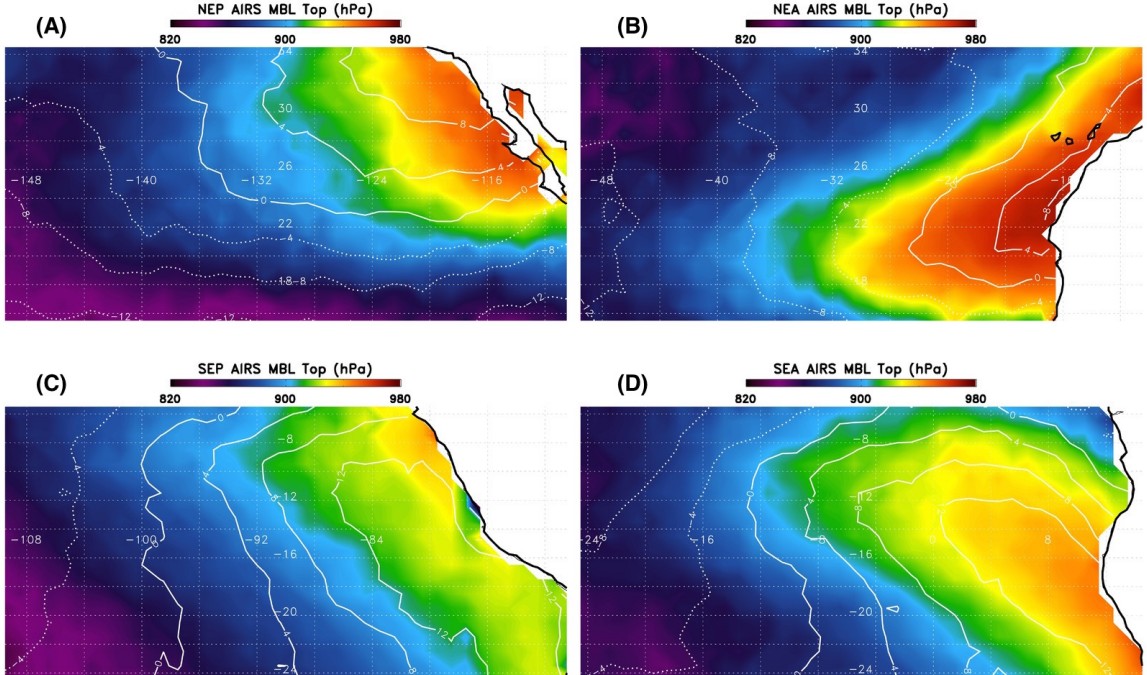

**Figure 4. MBL depth (hPa) for regions listed in Table 1: (a) NEP, (b) NEA, (c), SEP, and (d) SEA. The AIRS 1000-700 hPa dMSE is overlaid in white contors (solid are for positive and dashed for negative).**



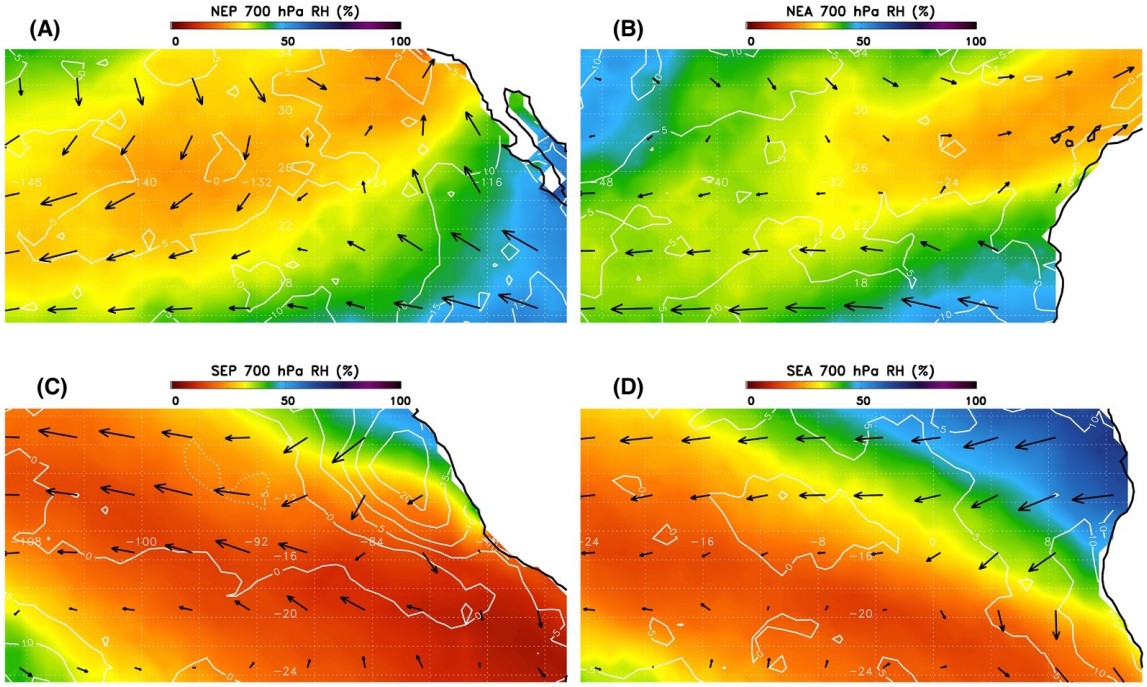

**Figure 5. AIRS RH$_{700}$ (%) for regions listed in Table 1: (a) NEP, (b) NEA, (c), SEP, and (d) SEA. The MERRA-AIRS RH$_{700}$ difference is shown as white contours (solid implies MERRA is moister, and dashed implies AIRS is moister). The length and direction of the arrows depict the 700 hPa wind vectors from MERRA.**





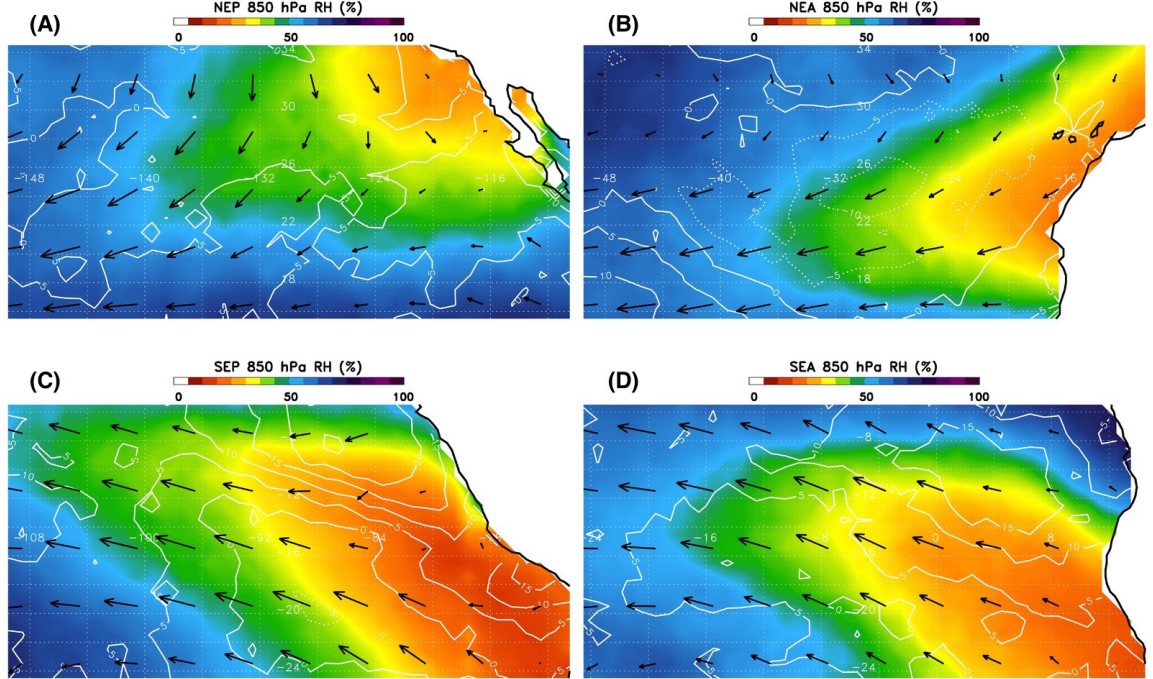

Figure 6. AIRS RH$_{850}$ (%) for regions listed in Table 1: (a) NEP, (b) NEA, (c), SEP, and (d) SEA. The MERRA-AIRS RH$_{850}$ difference is shown as white contours (solid implies MERRA is moister, and dashed implies AIRS is moister). The length and direction of the arrows depict the 850 hPa wind vectors from MERRA.




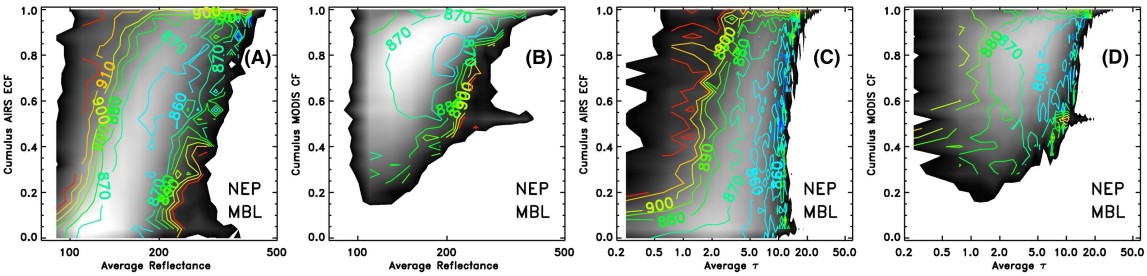

**Figure 7. Shown are joint pdfs for four different combinations of variables that are described in Section 4.2: (a) reflectance versus AIRS ECF, (b) MODIS τ versus AIRS ECF, (c) reflectance versus MODIS cloud fraction, and (d) MODIS τ versus MODIS cloud fraction. The black to grey to white scale is the natural log of total counts per bin, where black is alog(counts)=2.0 and white is alog(counts)=8.0. All values in the pdfs shown are for the cumulus regime. The color contours depict the MBL depth (hPa).**



**Figure 8. Joint pdfs of reflectance versus ECF for the four spatial regions listed in Table 1. The SEP in Fig.7 is repeated here for clarity.**

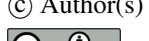



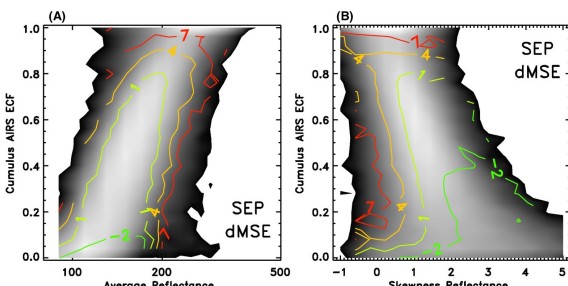

**Figure 9. Joint pdfs of reflectance average (left) and skewness (right) versus ECF for the SEP with dMSE depth as the overlay field. Other regions are very similar and are not shown for reasons of brevity.**



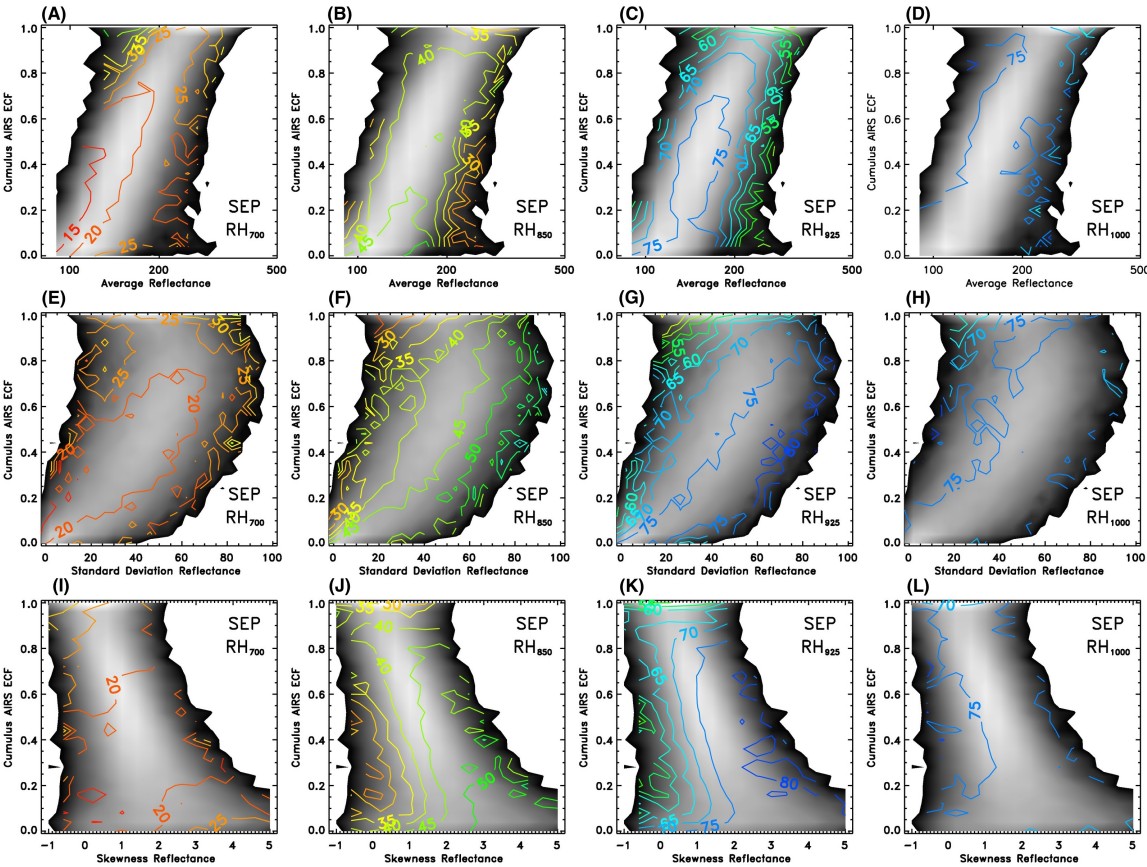

**Figure 10. Joint pdfs of mean reflectance versus ECF, with AIRS RH at 700, 850, 925, and 1000 hPa as overlays in the SEP for stratocumulus and cumulus.**





**Figure 11. Joint pdfs of reflectance skewness versus ECF for the four regions listed in Table 1, and the overlay field is $r_e$.**

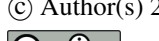



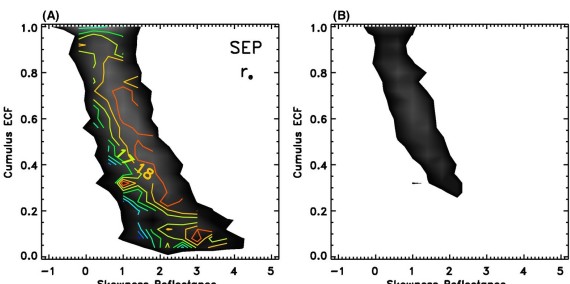

**Figure 12. (a) Same as Fig. 12n except sampling restricted to AMSU FORs that contain the CloudSat ground track. (b) Samples of the data in (a) that contain detected precipitation according to CloudSat.**



Figure 13. Joint pdfs of 700 hPa RH, theta, theta-e, u, and omega for the four regions listed in Table 1.





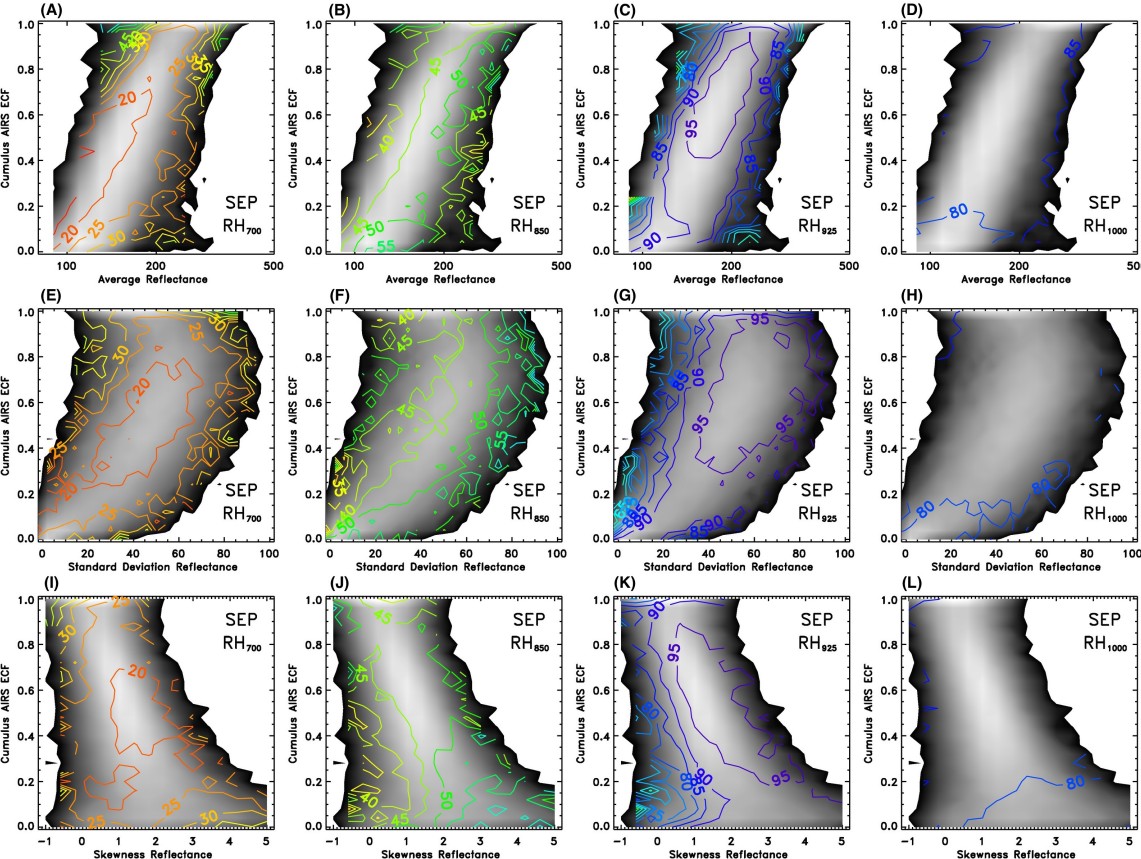

Figure A1. Same as Fig. 10 except for MERRA RH in the SEP region.