# Peer review of "A satellite and reanalysis view of cloud organization, thermodynamic, and dynamic variability within the subtropical marine boundary layer"

_Atmospheric Chemistry and Physics, 2017_

## Referee Comment (RC1) · Anonymous Referee #1 · 3 Mar 2017

This is a study of the relationships among cloud properties as retrieved by satellites and meteorological fields from MERRA. Most of the effort is in producing a dataset that combines AIRS/AMSU and MODIS data with MERRA at the smallest possible space and time scales. This effort is commendable and valuable, combining these data at small scales can potentially reveal a lot about the relationships between clouds and their environment. The analysis divides the data set into the four subtropical stratocumulus regions, though bigger than Klein-Hartmann regions to focus more on broken cloud regimes. Numerous quantities are examined, especially through the use of joint distributions and conditionally averaged quantities. This is mostly effective, but the weakness of the paper is that it meanders through the results without a lot of focus

which I think will lose a lot of readers. My main suggestion is to re-work section 4, but there are probably a couple of different ways that could be done. I will include some detailed comments, some of which might become irrelevant depending on how the manuscript changes in revision.

COMMENTS:

1. While the methodology overall seems very good, I have some concerns about sample sizes and statistics. The choice to only look at 2009 must be motivated by the effort expended to gather the raw data and process down to the combined data that is being used here. Very understandable, but it is not clear whether one season of the final combined data is enough to say much. This issue might be resolved with a few words about how many samples are actually retained. This comment definitely applies to the joint pdfs, too. Are the color bars different for the different regions, and how much data is in a black region compared to a white region?

2. Using three moments of the reflectance is interesting, but the physical interpretation gets lost in the text. It is made clear that skewness increases in cumulus regimes as ECF drops. Is the interpretation that this is a measure of cloud size? The standard deviation of reflectance seems to be connected to the boundary layer depth (Fig 8 & Page 9). Is that expected? The standard deviation isn't used much except to make this point, and it is not clear that it adds much to the overall story. Maybe it would be worth extracting the standard deviation of reflectance into supplemental material?

3. Section 4.2 is lacking. If I understand correctly, the point of this section is to whittle down the number of variables to look at in the later sections, settling on reflectance and ECF as the "phase space" (or maybe the "independent" or "predictor" variables?). The weakness is that the selection seems to be mostly arbitrary rather than by systematic evaluation. In the list of comparisons, two combinations are missing that would fill in the matrix: visible reflectance and tau; ECF and cloud fraction. Both of these seem like they would exhibit strong correlations, so maybe that is why they are omitted. But in other

parts of the text there is a contrast made between the MODIS and AIRS cloud fractions, so seeing ECF versus cloud fraction would be useful. That accounts for an assessment of the "phase space" variables, but MBL depth is also being discussed here, but it is not clear why. Is the MBL standing in here for an integrated "thermodynamic" variable?

4. The comparison of the regions. Early in the paper (Sections 1-3), it makes sense to look at the four regions separately. Going through Section 4, my feeling is that mostly the NEP, SEP, and SEA act very similarly, while NEA is an outlier. This is likely due to the NEA being more strongly influenced by midlatitude systems (even when filtered for mid and high clouds). A few points are raised about the difference between the hemispheres, but it isn't clear whether there is enough sampling (especially with only one season) to make any definitive statements. So I wondered, especially at the end of Section 4.3, whether it would simplify things to combine NEP, SEP, and SEA into one population and exclude NEA or show it as a contrasting population? The advantage is to reduce figure panels and increase overall sample size at the expense of having a comparison of the regions. In the present form, I don't see that the bottom line of the paper is really emphasizing any differences in the regions except that NEA is different from the others. As a related note, the title of Section 4.3 is "Regional differences in MBL depth and dMSE," but my main takeaway from Figure 8 is the similarity of the regions, and I felt like dMSE was not much emphasized in the section.

5. Comparing different scales. This study focuses on the smallest scales possible for the data, which is interesting by itself. There should be some care taken when comparing to previous studies that are explicitly working at much larger scales. This comes up in a few places in the text, but prominently at the end of section 4.3 where there is a conclusion that dMSE is correlated with small-scale spatial structure *rather than* large-scale thermodynamic structure. This might be misleading. When averaged up to longer time scales, it seems reasonable that dMSE is more representative of the large-scale thermodynamic structure than the spatial structure of clouds. The same holds for LTS and EIS; the relationships between these bulk measures of inversion strength and

cloud cover are only valid on relatively long time scales. Recall that the Klein-Hartmann line is derived using seasonal averages. This is discussed occasionally in the literature; one example is found in Zhang et al. (2009, DOI:10.1175/2009JCLI2891.1) where they point out that sampling the low-level divergence distribution is important for capturing the relationship between LTS and cloud cover.

6. Value of Section 4.4? The text seems to suggest that the point of this section is to compare AIRS and MERRA RH, showing they are similar and therefore useful. The MERRA RH isn't shown here (added as Figure A1), which undercuts this as the main message of the section. The section title is just "vertical structure of RH," but it is pretty hard to get a good sense for the vertical structure from the conditionally averaged contour plots showing one level at a time. The question is what aspect of the RH structure is needed to advance the overall argument of the paper? Based on Section 5, it is not clear that the vertical structure of RH is integral to the paper and Section 4.4 and Figure 10 could be deleted.

7. The connection to microphysical effects. Section 4.5 brings $r_e$ into the picture, and suggests that the difference between the stratocumulus and cumulus is due to microphysical processes. The next section makes the connection to wind speed, which is interesting. I'm not sure I understand the physical interpretation of the result. Also, it seems like making the link via the comparison of the contour plots in Figures 11 e-h and 13 e-h is a little cumbersome. Does viewing this relationship within the reflectance-cloud fraction phase space make the most sense here, and if so, what do we get from this view that would not appear by directly correlating $r_e$ and $u925$, for example? This seems like a key finding in the paper, and it might be better drawn out by combining sections 4.5 and 4.6 into a more unified discussion of the $r_e$ variation and connection to meteorology and microphysical processes.

---

## Referee Comment (RC2) · Anonymous Referee #2 · 10 Mar 2017

**"A satellite and reanalysis view of cloud organization, thermodynamic, and dynamic variability within the subtropical marine boundary layer"**
by Kahn, Brian H. et al.

**General comment and recommendation**

This manuscript presents a comparison of (correlations between) cloud properties and thermodynamic and dynamic fields derived from AIRS and MERRA data, with those derived in previous literature and derived in this paper from MODIS. The manuscript comes across as rather unfocused, wandering between a variety of objectives, none of which end up convincingly presented. The manuscript appears to: a) evaluate AIRS and MERRA against MODIS and other cited products; b) provide physical insights into what explains the transition by means of reflectance, optical depth, boundary layer depth and effective radius; c) present the AIRS cloud product that can best reflect the transition from stratocumulus to cumulus, where a variety of measures are tried out; d) present different physical behaviours between four regions in which the transition between stratocumulus and cumulus occurs.

Sometimes the lack of focus and a specific question of interest seems to shines through in the authors' writing, for instance, when they introduce new sections, which might represent choices that are "not optimal", but simply provide "a fresh look at available products", or when they describe that choosing which variables to plot in their joint pdfs is challenging. If the focus would be on presenting novel insights, I had expected that beyond abstract descriptions of behaviour of different quantities in the joint pdf's the authors explain what this behaviour actually tells us about observed cloud fields. If the focus would be on an evaluation of AIRS products, I would expect that the evaluation were more thorough and go beyond a comparison of seasonal averaged fields. What contributes to the wandering is that the authors use different data sets for different objectives as they present themselves. AIRS and MERRA are used for interpreting skewness measures and the transition between cloud types, along with comparisons of MODIS. Only MODIS is used for the purpose of evaluating effective radii in the two regimes, and what might physically or methodologically explain effective radii behaviour. In much of the authors assertions, previous literature is referenced, but often not explained.

Because the manuscript does not present a novel insight and fails to convincingly argue for any method or the AIRS or MERRA datasets at providing novel insights, I recommend a rejection of the manuscript.

**Specific comments**

- The title is unspecific (which satellite and reanalysis data sets?) and promises more insight than the paper offers. The term cloud organization is not well-chosen, because the authors do not present results on cloud organization nor discuss what cloud organization means. The words organised and disorganised are repeatedly used throughout the manuscript, but mostly in reference to *organised* stratocumulus and *disorganised* trade-wind cumulus. Both stratocumulus and trade-cumulus can be organized and disorganized, depending on some definition of organization. Sometimes it seems the authors refer to homogeneous and heterogeneous, but mostly it seems that they refer to the two different cloud types.

- The word novel is repeatedly used, but seems an overstatement. In much of the manuscript, the authors confirm insights found in previous studies, and that citation list is long.

- One novelty that is argued for is the use of AIRS and MERRA datasets at their instantaneous native resolution. But to prove the suitability for these datasets for this kind of study, the authors qualitatively compare the morphology of the stratocumulus to cumulus transition from seasonal averaged AIRS and MERRA data with the morphology known from existing studies. I do not think a qualitative comparison of the seasonal mean transition tells us enough about how good AIRS and MERRA perform at their native resolution.

- The authors make an argument for separating the cloud regimes stratocumulus and cumulus based on infrared-based thermodynamic phase (rather than by dynamical regime such as done in previous literature). The thermodynamic phase provides information about whether just liquid or ice is present in the detected clouds. Based on a single scene in Figure 1 and 2 the authors argue that stratocumulus is well identified by those pixels that are detected as liquid, whereas trade-wind cumulus are those pixels that have an unknown thermodynamic phase. How do the authors know that this separation holds well for other scenes? After all, trade-wind cumulus are also made of liquid only, and it is unclear and not explained why they could not be identified as such in other scenes.

  It is also not clear for what purpose the two cloud types are separated here in this paper. Mostly this seems to be a proposition to use AIRS thermodynamic phase in future studies, but with insufficient evidence.

- One aspect of the paper that prevents it from providing clear physical insights (if this were the main objective) is that the authors never explain what the skewness in reflectance or optical depth tells us about the nature of the cloud field that is observed (and this is true for many of the behaviours derived from the joint pdfs). The skewness measure has been used in previous studies, and can with some background of course be interpreted, but the authors never explicitly do. This makes the description of results rather abstract.

- The discussion in section 4.5 and the conclusions argue for both physical causes (precipitation) as well as retrieval-related biases (inhomogeneity) for the observed larger effective radius in cumulus clouds compared to stratocumulus. But whereas first is stated that (L24) "the observed increase in re is entirely consistent with environmental variability (winds/droplet growth/precipitation)", it is written further along that the greater inhomogeneity in such precipitating cumulus fields can cause assumptions used in retrievals to break down. Hence, should I trust the retrieved larger effective radii observed?

- In the last paragraphs of section 4.6 and the summary, the authors argue two seemingly contradicting statements with which they end their manuscript. Namely, that three of the four regions studied show similar relationships and behaviours among cloud-related quantities and the (thermo)dynamic state, but also that the relationships are non-unique (can vary greatly), for which their datasets provide a good opportunity for further exploration. I understand the subtlety, but is this the best ending?

---

## Author Comment (AC1) · 30 Mar 2017

**The authors appreciate the encouraging, thoughtful and helpful comments and suggestions regarding the manuscript by the reviewer, and appreciate the time and effort spent on it.**

This is a study of the relationships among cloud properties as retrieved by satellites and meteorological fields from MERRA. Most of the effort is in producing a dataset that combines AIRS/AMSU and MODIS data with MERRA at the smallest possible space and time scales. This effort is commendable and valuable, combining these data at small scales can potentially reveal a lot about the relationships between clouds and their environment. The analysis divides the data set into the four subtropical stratocumulus regions, though bigger than Klein-Hartmann regions to focus more on broken cloud regimes. Numerous quantities are examined, especially through the use of joint distributions and conditionally averaged quantities. This is mostly effective, but the weakness of the paper is that it meanders through the results without a lot of focus which I think will lose a lot of readers.

**Since reviewer #2 had similar thoughts on the lack of focus in section 4, we will carefully reorganize the flow of the paper per the suggestions of reviewer #1 below.**

My main suggestion is to re-work section 4, but there are probably a couple of different ways that could be done. I will include some detailed comments, some of which might become irrelevant depending on how the manuscript changes in revision.

**We will make some major changes to section 4 as suggested by the reviewer below in the detailed comments.**

COMMENTS:

1. While the methodology overall seems very good, I have some concerns about sample sizes and statistics. The choice to only look at 2009 must be motivated by the effort expended to gather the raw data and process down to the combined data that is being used here. Very understandable, but it is not clear whether one season of the final combined data is enough to say much. This issue might be resolved with a few words about how many samples are actually retained. This comment definitely applies to the joint pdfs, too. Are the color bars different for the different regions, and how much data is in a black region compared to a white region?

**This is a good point.  We were indeed limited by the sheer volume of data and processing required. We have processed the full year of 2009 for the entire globe. While we initially debated about presenting the full 2009 record, we felt as though the seasonal story would get lost in the story about the four regions. We also could not decide on a simple set of figures, bar charts, or tables that might show the seasonal variability (that could potentially quadruple, or more, the figures and panels).  Also, we note that the seasonal variability is sensitive to the latitude and**

longitude of the region selected.  For instance, in the SEP region of study, during DJF (local summer), convection and moist intrusions impede on the northern side of the region and that causes systematic changes in portions of the joint pdfs.  We concluded that we stood on the most firm ground by choosing JJA in the NH and SON in the SH that correspond with peak cloud frequency as shown in Klein and Hartmann (1993).

In the revision we will add a table that shows the raw pixel counts (or grid counts for MERRA) for all data that go into the joint pdfs.

The gray scale of the joint pdfs that show the counts was settled on after many revisions and ideas.  It indicates the log(count), where black is log(3) and white goes to log(8) or so.  We will add a single gray scale bar at the bottom of every joint pdf figure for clarity.

2. Using three moments of the reflectance is interesting, but the physical interpretation gets lost in the text. It is made clear that skewness increases in cumulus regimes as ECF drops. Is the interpretation that this is a measure of cloud size? The standard deviation of reflectance seems to be connected to the boundary layer depth (Fig 8 & Page 9). Is that expected? The standard deviation isn't used much except to make this point, and it is not clear that it adds much to the overall story. Maybe it would be worth extracting the standard deviation of reflectance into supplemental material?

We lost reviewer #2 in regards to the connection (poorly) made between cloud organization and skewness.  We will revise the discussion of the moments accordingly in the revision and will be clearer about how they connect to the other variables.

With regard to the interpretation of reduced ECF because of smaller cloud size, that is a great question but we cannot provide a firm answer.  There could be several causes.  First, if the cloud opacity is reduced, the ECF will go down even though the cloud coverage remains constant over the entire AIRS pixel. That is because the ECF is a convolution of emissivity and cloud fraction.  Second, it is possible that the emissivity remains fixed but the cloud coverage becomes more broken, also reducing ECF.  Third, if the ECF is further reduced (increased), it could be that the small cloud elements could be more widely spaced (packed together) even though the cloud size may be the same. Fourth, it is likely a combination of the first three factors in different combinations depending on the cloud scene and time period.

With regard to the standard deviation, in the few uncommon cases with very high standard deviation (note the blacker shading), that the MBL is quite a bit deeper than in the mean and skewness dimensions.  These cases are aligned with the highest values of RH in the standard deviation seen in Figure 10 for the SEP. We agree that the standard deviation was not sufficiently teased out in the text.  We can speculate about the causes, but in lieu of careful investigation beyond the scope of this work, probably the best approach is to move the standard deviations in Figure 8 to the

**Appendix.**

3. Section 4.2 is lacking. If I understand correctly, the point of this section is to whittle down the number of variables to look at in the later sections, settling on reflectance and ECF as the "phase space" (or maybe the "independent" or "predictor" variables?). The weakness is that the selection seems to be mostly arbitrary rather than by systematic evaluation. In the list of comparisons, two combinations are missing that would fill in the matrix: visible reflectance and tau; ECF and cloud fraction. Both of these seem like they would exhibit strong correlations, so maybe that is why they are omitted. But in other parts of the text there is a contrast made between the MODIS and AIRS cloud fractions, so seeing ECF versus cloud fraction would be useful. That accounts for an assessment of the "phase space" variables, but MBL depth is also being discussed here, but it is not clear why. Is the MBL standing in here for an integrated "thermodynamic" variable?

**We agree that the starting point for dimensionality choice is pretty arbitrary. We draw upon a response to reviewer #2 to partially address this concern:**

**"The most honest way to go into this investigation is to ask what do we do when confronted with a choice from an enormous selection of available data? Dozens of geophysical variables are available from each instrument and reanalysis system. These variables can be plotted against each other in 1000s of combinations (or much more). The moments of these variables are also another dimensional choice, so to speak. On top of that, any field can be overlaid onto these two dimensions as done in the figures. So where does one start? As we pointed out, our reasoning for starting where we did is found here:**

**Line 3, page 8: "Motivated in large part to link cloud and thermodynamic properties derived from infrared and visible bands…"**

**For the revision, it is a great idea to add the two additional panels for visible reflectance and tau, and ECF and cloud fraction. We will do this.**

**Lastly, we chose MBL depth as a representative variable of the MBL just to make the point of why we chose the dimensions that we did. We have made these plots with RH, dMSE, etc., and the story is very similar. Basically, the overlying quantity in the joint pdf has a larger dynamic range with the dimensions when choosing ECF and reflectance. The values are distributed more widely across the dimensions and look more structured. The optical thickness retrievals from MODIS are only obtained for a subset of all MBL clouds since the retrievals fail within cumulus that are subpixel in size. This is why we made the argument in the paper that there is a population of clouds at the subpixel scale picked up in the AIRS reflectance data that is completely missed by the MODIS cloud mask and cloud optical property retrievals, driving our choice for reflectance in the end. We will tease this out more fully in the revision.**

4. The comparison of the regions. Early in the paper (Sections 1-3), it makes sense to

look at the four regions separately. Going through Section 4, my feeling is that mostly the NEP, SEP, and SEA act very similarly, while NEA is an outlier. This is likely due to the NEA being more strongly influenced by midlatitude systems (even when filtered for mid and high clouds). A few points are raised about the difference between the hemispheres, but it isn't clear whether there is enough sampling (especially with only one season) to make any definitive statements. So I wondered, especially at the end of Section 4.3, whether it would simplify things to combine NEP, SEP, and SEA into one population and exclude NEA or show it as a contrasting population? The advantage is to reduce figure panels and increase overall sample size at the expense of having a comparison of the regions. In the present form, I don't see that the bottom line of the paper is really emphasizing any differences in the regions except that NEA is different from the others. As a related note, the title of Section 4.3 is "Regional differences in MBL depth and dMSE," but my main takeaway from Figure 8 is the similarity of the regions, and I felt like dMSE was not much emphasized in the section.

**Thanks for the suggestion of combining the three regions. The sample sizes are quite large for portions of the joint pdfs that are gray-ish and not as large for other portions of the pdfs that are black-ish (see figure 8, upper row). Some differences still show themselves throughout the paper even in portions of the pdfs where the sample sizes are larger, so we would be concerned that those differences would be averaged out into a composite pdf that more poorly resembles each of the three regions individually. Another significant concern is that the distribution of samples throughout the joint pdf can be different in the SEP, NEP, and SEA that would further smear out the subtle differences if all were summed together into a single pdf.**

**We appreciate that the subtle differences among the three regions may arise because of insufficient sample size, or because there is year-to-year variability and the differences can be easily flipped around to another year. While this may be true to some degree, there are some subtle differences that we believe are actual differences between the regimes in portions of the pdfs, specifically in relation to MBL depth, dMSE, omega, reff, and u925. We will emphasize these more subtle behaviors in the revised text. We may consider highlighting certain portions of the pdfs in the subpanels with boxes or labels or lettering to point these features out.**

**In the revision, we will also make clear that the overall similarity is the most apparent feature. Lastly, we will think about an improved title for the subsection.**

**We will move the standard deviation panels to the Appendix for figures 8 and 11 per the previous response above.**

5. Comparing different scales. This study focuses on the smallest scales possible for the data, which is interesting by itself. There should be some care taken when comparing to previous studies that are explicitly working at much larger scales. This comes up in a few places in the text, but prominently at the end of section 4.3 where there is a conclusion that dMSE is correlated with small-scale spatial structure *rather than* large-scale

thermodynamic structure. This might be misleading. When averaged up to longer time scales, it seems reasonable that dMSE is more representative of the large scale thermodynamic structure than the spatial structure of clouds. The same holds for LTS and EIS; the relationships between these bulk measures of inversion strength and cloud cover are only valid on relatively long time scales. Recall that the Klein-Hartmann line is derived using seasonal averages. This is discussed occasionally in the literature; one example is found in Zhang et al. (2009, DOI:10.1175/2009JCLI2891.1) where they point out that sampling the low-level divergence distribution is important for capturing the relationship between LTS and cloud cover.

**Thanks for pointing out the scale context of the agreement. In the revision, we will be clearer about this and will revisit our references and text with regard to extrapolating between small land large scales, and instantaneous versus seasonal time periods. We will make the point that there is added value in quantifying instantaneous matchups with dMSE and cloud structure, but that does not negate its correspondence with the large scale thermodynamic state.**

6. Value of Section 4.4? The text seems to suggest that the point of this section is to compare AIRS and MERRA RH, showing they are similar and therefore useful. The MERRA RH isn't shown here (added as Figure A1), which undercuts this as the main message of the section. The section title is just "vertical structure of RH," but it is pretty hard to get a good sense for the vertical structure from the conditionally averaged contour plots showing one level at a time. The question is what aspect of the RH structure is needed to advance the overall argument of the paper? Based on Section 5, it is not clear that the vertical structure of RH is integral to the paper and Section 4.4 and Figure 10 could be deleted.

**We agree with the reviewer that this section is somewhat tangential and in the revision we will move Fig. 10 to the Appendix. We will absorb some of the text into the Appendix as well but will try and delete a good portion of it. We would like to keep the MERRA and AIRS RH figures in the Appendix because it shows very clearly the moistening and drying with respect to reflectance and ECF, and that RH does not simply depend on altitude. These points will be made clear in the revised text.**

7. The connection to microphysical effects. Section 4.5 brings r_e into the picture, and suggests that the difference between the stratocumulus and cumulus is due to microphysical processes. The next section makes the connection to wind speed, which is interesting. I'm not sure I understand the physical interpretation of the result. Also, it seems like making the link via the comparison of the contour plots in Figures 11 e-h and 13 e-h is a little cumbersome. Does viewing this relationship within the reflectance cloud fraction phase space make the most sense here, and if so, what do we get from this view that would not appear by directly correlating r_e and u925, for example? This seems like a key finding in the paper, and it might be better drawn out by combining sections 4.5 and 4.6 into a more unified discussion of the r_e variation and connection to meteorology and microphysical processes.

The physical connection between wind speed and effective radius is found in the first paragraph of Section 4.6 on page 11 and is motivated by the work of Nuijens et al. (2009) and follow-on studies based on bulk theory afterwards.

Given that we will remove the standard deviations in figure 11 in the revision and place them into the Appendix, we can move the wind speed panels from figure 13 to figure 11 as the lower row for easier comparison.

Given that we have selected the reflectance dimension as the most appropriate for comparison, we will leave that dimension as is for the revision. However, we will add correlations between u925 and effective radius for the revision as a new figure. Part of the reason we wanted reflectance for effective radius is that it is easier to show some of the 3-D radiative transfer issues that arise in the highly skewed portions of the joint pdfs that are discussed in section 4.4.

Given these comments above and elsewhere, we will combine section 4.5 and 4.6 into a new unified section that united effective radius, u925, omega, theta and thetae.

---

## Author Comment (AC2) · 30 Mar 2017

**We thank the reviewer for taking the time and effort to review the manuscript and appreciate the comments. In this response, we aim to highlight more clearly the purpose, value, and novelty claimed in the manuscript.**

General comment and recommendation

This manuscript presents a comparison of (correlations between) cloud properties and thermodynamic and dynamic fields derived from AIRS and MERRA data, with those derived in previous literature and derived in this paper from MODIS.

**This paper is clearly not a comparison or a validation paper. Its purpose is to describe the synergistic use of previously validated data products from AIRS, MODIS, and CloudSat, together with MERRA reanalysis at the native temporal and spatial resolution, to investigate relationships between cloud microphysical and optical properties, and dynamical and thermodynamic fields. To our knowledge, at the time of submission, no attempt of this kind of approach has been made for investigating the marine boundary layer.**

The manuscript comes across as rather unfocused, wandering between a variety of objectives, none of which end up convincingly presented.

**This is a fair statement. We agree that we can tighten up the organization of the various components in the manuscript, and be more forthright with our conclusions and take home messages.**

The manuscript appears to: a) evaluate AIRS and MERRA against MODIS and other cited products;

**As stated above, this paper is clearly not a comparison or a validation paper. In fact, the only common field between the three instruments (AIRS, MODIS, CloudSat) and one reanalysis (MERRA) used in the manuscript is the vertical structure of relative humidity (Section 4.4 and Appendix A), which we will move to the Appendix in the revision per reviewer #1's suggestion. AIRS, MODIS, CloudSat, and MERRA each provide unique information that can be brought to bear on observing the subtropical MBL. We are not comparing common geophysical fields obtained between them. Our guiding philosophy is "Why not play to the strengths of each instrument?"**

b) provide physical insights into what explains the transition by means of reflectance, optical depth, boundary layer depth and effective radius;

**That is (partially) correct, although there are many other geophysical variables used, and various moments (mean, variance, skewness) that highlight certain aspects of MBL structure. However, we are not attempting to "explain the**

**transition" so much as rather present a new way to observe it.**

c) present the AIRS cloud product that can best reflect the transition from stratocumulus to cumulus, where a variety of measures are tried out;

**Yes.  There are three cloud products in particular from AIRS that are used. (1) The AIRS cloud thermodynamic phase product is used to coarsely group together uniform closed cellular stratocumulus and broken, disorganized open cellular trade cumulus clouds. (2) The AIRS effective cloud fraction (ECF) is derived from infrared channels so it will have a different perspective of cloud cover compared to visible reflectance or optical thickness. (3) The AIRS visible channels are used to quantify the reflectance that is filtered by an AIRS visible cloud mask.**

d) present different physical behaviours between four regions in which the transition between stratocumulus and cumulus occurs.

**As this work progressed, it became apparent that the data revealed that each of the four regions has some subtle differences when contrasted against the other regions, but the NEA is the biggest outlier of the four.  These differences might be reflected in dMSE, vertical velocity, liquid water cloud effective radius, so on and so forth. We did not attempt to explain why the four regions show such differences, which is well beyond the scope of this investigation and would undoubtedly require extensive numerical modeling experiments, and further investigation of several years of data.**

Sometimes the lack of focus and a specific question of interest seems to shines through in the authors' writing, for instance, when they introduce new sections, which might represent choices that are "not optimal", but simply provide "a fresh look at available products",

**It appears the reviewer is referring to line 13, page 8 for the quoted text.  We offered specific reasons for why we chose the effective cloud fraction (ECF) variable from AIRS over the cloud fraction (CF) derived from the MODIS cloud mask, and also reflectance from AIRS over the optical thickness from MODIS.  The reasons are described just above the quoted text:**

**lines 7-11, page 8: "The MBL depth exhibits clearer patterns in the ECF dimension rather than the cloud fraction dimension. The latter is more compressed and the gradients are weaker in both dimensions. The MBL depth is deepest for lower values of ECF, $\tau$, and reflectance. In addition, the MBL depth also decreases for the most reflective clouds at a given value of ECF while this behavior is not observed for $\tau$. We posit that an additional population of sub-pixel cumulus clouds is captured within the reflectance data that is not captured in $\tau$ data."**

or when they describe that choosing which variables to plot in their joint pdfs is challenging.

**It appears the reviewer is referring to line 2, page 8. The most honest way to go into this investigation is to ask what do we do when confronted with a choice from an enormous selection of available data? Dozens of geophysical variables are available from each instrument and reanalysis system. These variables can be plotted against each other in 1000s of combinations (or much more). The moments of these variables are also another dimensional choice, so to speak. On top of that, any field can be overlaid onto the two dimensions as done throughout the joint pdf figures. So where does one start? As we pointed out, our reasoning for starting where we did is found here:**

**Line 3, page 8: "Motivated in large part to link cloud and thermodynamic properties derived from infrared and visible bands…"**

If the focus would be on presenting novel insights,

**Since the reviewer is emphasizing multiple times the "novelty" of this work, after doing a word search we found only three instances of the word "novel" are used in the manuscript. Perhaps this word choice is unfortunate and we will remove accordingly in the revision.**

I had expected that beyond abstract descriptions of behavior of different quantities in the joint pdf's the authors explain what this behaviour actually tells us about observed cloud fields.

**The paper is not only about observed cloud fields. It is an attempt to describe a more holistic synthesis of the subtropical MBL from the point of view of A-train satellite observations and MERRA reanalysis built from native temporal and spatial resolution data. This includes winds, T/q/RH structure, the occurrence of light precipitation, vertical and horizontal motion, the depth of the MBL, and how they link to cloud properties. This paper is not about the physical causes of the stratocumulus to cumulus transition, but we do cite some of these papers in the Introduction.**

If the focus would be on an evaluation of AIRS products, I would expect that the evaluation were more thorough and go beyond a comparison of seasonal averaged fields.

**As stated above, this paper is clearly not a comparison, validation, or data evaluation paper for AIRS products. We have cited references throughout that point the reader to previous validation efforts that support the use of the data as shown in the manuscript.**

What contributes to the wandering is that the authors use different data sets for different objectives as they present themselves.

**The whole purpose of the paper is to use the different instruments and reanalysis**

**data sets as building blocks to construct a simultaneous point of view of the MBL, playing on the strengths of each instrument. The reviewer comment strongly suggests we can be much more clear and concise about our purpose. We will revise accordingly.**

AIRS and MERRA are used for interpreting skewness measures and the transition between cloud types, along with comparisons of MODIS. Only MODIS is used for the purpose of evaluating effective radii in the two regimes, and what might physically or methodologically explain effective radii behaviour.

**As far as we know, MODIS has the most useful, validated, tested, and investigated global retrieval of liquid water cloud effective radius available to the scientific community. AIRS does not provide one. CloudSat uses MODIS effective radius in its forward algorithm of retrieval products. MERRA is quite awful at clouds. MODIS is, for all practical purposes, the only ballgame in town.**

In much of the authors assertions, previous literature is referenced, but often not explained.

**We tried to be as comprehensive as possible with citing references for our statements. In the revision we will try and be as clear as possible as to why we are citing a particular work.**

Because the manuscript does not present a novel insight and fails to convincingly argue for any method or the AIRS or MERRA datasets at providing novel insights, I recommend a rejection of the manuscript.

**We would like to bring up that reviewer #1 had a different opinion: "This effort is commendable and valuable, combining these data at small scales can potentially reveal a lot about the relationships between clouds and their environment."**

**We hope that we will convince reviewer #2 the value of this work pending the revisions that we will make.**

Specific comments

The title is unspecific (which satellite and reanalysis data sets?) and promises more insight than the paper offers.

**There are plenty of papers that have general titles when there are numerous data sets described because the titles would be too long. (Also, for example, it is common that papers that use the CMIP archive do not reference particular models used.) The words "A satellite and reanalysis view" suggests that this is an observational study rather than a study that deduces the complex physical mechanisms at play in the MBL. We could consider changing it to "An A-train and MERRA view" if that helps.**

The term cloud organization is not well-chosen, because the authors do not present results on cloud organization nor discuss what cloud organization means. The words organised and disorganised are repeatedly used throughout the manuscript, but mostly in reference to organised stratocumulus and disorganised trade-wind cumulus. Both stratocumulus and trade-cumulus can be organized and disorganized, depending on some definition of organization. Sometimes it seems the authors refer to homogeneous and heterogeneous, but mostly it seems that they refer to the two different cloud types.

**In our revision we will be more concise. The particular choice of words was meant to follow on to the work of Muhlbauer et al. When we refer to cloud organization in the revision, we will be more specific and cite values of skewness rather than used 'organized' or 'disorganized'.**

The word novel is repeatedly used, but seems an overstatement. In much of the manuscript, the authors confirm insights found in previous studies, and that citation list is long.

**We found the word 'novel' was used only three times in the entire manuscript. We will select another word or rephrase accordingly.**

**We feel as though it is a particular strength of this approach that we are able to reaffirm a long list of previous findings and reference a large body of research.**

One novelty that is argued for is the use of AIRS and MERRA datasets at their instantaneous native resolution. But to prove the suitability for these datasets for this kind of study, the authors qualitatively compare the morphology of the stratocumulus to cumulus transition from seasonal averaged AIRS and MERRA data with the morphology known from existing studies. I do not think a qualitative comparison of the seasonal mean transition tells us enough about how good AIRS and MERRA perform at their native resolution.

**The seasonal averages were developed as a first order check on our methods and use of data. Since the seasonal averages agreed very well with previous research, that gave us confidence in moving forward with the joint pdfs. (Also, it was one of the only ways to compare with previous research since so few studies have looked at joint pdfs in the manner that we showed in figures 7-13.) In the revision we will be clear about why we start with the seasonal averages then proceed to the joint pdfs.**

The authors make an argument for separating the cloud regimes stratocumulus and cumulus based on infrared-based thermodynamic phase (rather than by dynamical regime such as done in previous literature).

**The motivation for this approach is found in the Methodology section. As this is a pixel based approach, we require that all ice cloud instances are removed, and we are confident that the AIRS phase product is more than sufficient.**

**Page 5, lines 6-7: "Removal of pixels containing mid- and high-level clouds helps to reduce ambiguities introduced by free tropospheric clouds and also a portion of the thermodynamic and dynamic variability associated with cloudy areas of synoptic-scale waves."**

**The dynamical approach is consistent with this approach in the sense that stratocumulus clouds show larger free tropospheric subsidence than the cumulus clouds. We will revise the manuscript accordingly to emphasize these points.**

The thermodynamic phase provides information about whether just liquid or ice is present in the detected clouds.

**We refer the reviewer to page 5, lines 11-12: "Jin and Nasiri (2014) showed that AIRS successfully identifies the presence of ice within the AIRS FOV in excess of 90% of the time when compared to CALIPSO thermodynamic phase estimates."**

**AIRS is an extremely radiometrically stable instrument with very strong sensitivity to cloud phase as discussed in Kahn et al. (2014) and Jin and Nasiri (2014) and citations within.**

Based on a single scene in Figure 1 and 2 the authors argue that stratocumulus is well identified by those pixels that are detected as liquid, whereas trade-wind cumulus are those pixels that have an unknown thermodynamic phase. How do the authors know that this separation holds well for other scenes?

**Please refer to above response. We have evaluated AIRS against CALIOP (Jin and Nasiri, 2014), and Kahn et al. (2015), J. Geophys. Res. in the case of MODIS phase. These evaluations were performed globally for large sets of observations.**

After all, trade-wind cumulus are also made of liquid only, and it is unclear and not explained why they could not be identified as such in other scenes. It is also not clear for what purpose the two cloud types are separated here in this paper. Mostly this seems to be a proposition to use AIRS thermodynamic phase in future studies, but with insufficient evidence.

**We agree that the delineation between stratocumulus (liquid) and cumulus (unknown) should be made clearer in the revision. Since the AIRS cloud phase is based on channel selection that exploits the differences in the index of refraction for liquid and ice, if the cloud amount in the AIRS pixel is small enough the spectral signature will be so small that it does not trigger a liquid test (see Jin and Nasiri, 2014). We do know that there is cloud in the pixel using the ECF field (validated using CALIPSO lidar, see Kahn et al., 2014), so what is happening is that none of the phase tests are triggered even though a small amount of cloud is there. These cases line up very well with trade cumulus in the four regions selected.**

One aspect of the paper that prevents it from providing clear physical insights (if this were the main objective) is that the authors never explain what the skewness in reflectance or optical depth tells us about the nature of the cloud field that is observed (and this is true for many of the behaviours derived from the joint pdfs). The skewness measure has been used in previous studies, and can with some background of course be interpreted, but the authors never explicitly do. This makes the description of results rather abstract.

**We will strengthen this aspect of the interpretation of the data. On the whole, the more skewed the reflectance is, the smaller the ECF is. When the reflectance is approximately Gaussian, the ECF is larger. The former is seen very clearly in the cumulus pdfs and the latter in the stratocumulus pdfs. Since there is such good separation between the two cloud types, they should be discussed separately. (This also should be considered as an independent confirmation of the sensitivity of the AIRS phase algorithm to cloud type.) Even for the same combination of reflectance and ECF in cumulus and stratocumulus pdfs for the MBL depth, the MBL depth is shallower for stratocumulus. The same is true for dMSE (more positive for stratocumulus than trade cumulus.) This is a really interesting result that shows there is cloud regime dependence even for the same value of ECF and reflectance, and that separation is facilitated by the AIRS phase algorithm categories liquid and unknown. We will revise the text accordingly to strengthen these discussion points.**

The discussion in section 4.5 and the conclusions argue for both physical causes (precipitation) as well as retrieval-related biases (inhomogeneity) for the observed larger effective radius in cumulus clouds compared to stratocumulus. But whereas first is stated that (L24) "the observed increase in re is entirely consistent with environmental variability (winds/droplet growth/precipitation)", it is written further along that the greater inhomogeneity in such precipitating cumulus fields can cause assumptions used in retrievals to break down. Hence, should I trust the retrieved larger effective radii observed?

**On lines 16-18 on the same page we state the following: "As these particular MODIS pixels are limited to successful retrievals only, we offer evidence that the increase in re is entirely consistent with environmental variability that is furthermore consistent with droplet growth and precipitation." Since precipitating retrievals might be more inhomogeneous than non-precipitating ones, that alone could be a cause of the increase in re using the MODIS look up table approach. We did not claim otherwise. We simply showed that these larger values of effective radius strongly correspond to occurrences of precipitation detected by CloudSat.**

**Given the comments of reviewers #1 and #2, we will investigate this further for the revision. We will attempt to map an inhomogeneity parameter onto the retrievals of effective radius that are both precipitating and not precipitating, for a range of wind speeds.**

In the last paragraphs of section 4.6 and the summary, the authors argue two seemingly

contradicting statements with which they end their manuscript. Namely, that three of the four regions studied show similar relationships and behaviours among cloud-related quantities and the (thermo)dynamic state, but also that the relationships are non-unique (can vary greatly), for which their datasets provide a good opportunity for further exploration. I understand the subtlety, but is this the best ending?

**Good point. This ending needs some work and we will revise accordingly for the revision.**

---

## Author Response (AR1)

**The authors appreciate the reviewer's encouraging, thoughtful and helpful comments and suggestions regarding the manuscript, and appreciate the time and effort spent on it.**

**For this particular final response, we have added specific references to line and page numbers with respect to the track change version, figure and table changes, and additional comments on top of those provided during the discussions phase. In the track change version, all deletions are marked with a  and all additions are marked in red.**

**Note that we have changed 'reflectance' to 'radiance' throughout the manuscript and use the latter in the responses to be consistent with the units reported in the AIRS visible Level 1 files (Watts/meter\*\*2/micron/steradian).**

This is a study of the relationships among cloud properties as retrieved by satellites and meteorological fields from MERRA. Most of the effort is in producing a dataset that combines AIRS/AMSU and MODIS data with MERRA at the smallest possible space and time scales. This effort is commendable and valuable, combining these data at small scales can potentially reveal a lot about the relationships between clouds and their environment. The analysis divides the data set into the four subtropical stratocumulus regions, though bigger than Klein-Hartmann regions to focus more on broken cloud regimes. Numerous quantities are examined, especially through the use of joint distributions and conditionally averaged quantities. This is mostly effective, but the weakness of the paper is that it meanders through the results without a lot of focus which I think will lose a lot of readers.

**Reviewer #2 also had similar concerns. Below we describe the changes made in response to the reviewer comments.**

My main suggestion is to re-work section 4, but there are probably a couple of different ways that could be done. I will include some detailed comments, some of which might become irrelevant depending on how the manuscript changes in revision.

COMMENTS:

1. While the methodology overall seems very good, I have some concerns about sample sizes and statistics. The choice to only look at 2009 must be motivated by the effort expended to gather the raw data and process down to the combined data that is being used here. Very understandable, but it is not clear whether one season of the final combined data is enough to say much.

**We were indeed limited by the sheer volume of data and processing required. The full year of 2009 is processed for the entire globe. While the authors debated about presenting the full 2009 record, we concluded that the seasonal story would get lost**

within the story about the four regions. Furthermore, a simple set of figures, bar charts, or tables that might show the seasonal variability could not be decided on. Also, the seasonal variability is very much sensitive to the latitude and longitude of the region selected.  For instance, in the SEP region of study, during DJF (local summer), convection and tropical moist intrusions impede on the northern side of the regional box that creates systematic changes in parts of the joint pdfs.  By choosing JJA in the NH and SON in the SH that correspond with peak cloud frequency as shown in Klein and Hartmann (1993), we felt we stood on the most firm ground for this investigation.

This issue might be resolved with a few words about how many samples are actually retained.

**An additional column is added to Table 1 that shows the total number of data points used at the AIRS/AMSU field of regard spatial scale.  Although there are slight differences in the counts between the four regions, the total number varies only slightly between ~180,000 and ~186,000 during the three-month period listed in Table 1.**

**We have also added the following text to the manuscript on p. 7, lines 9-11: "*The total number of collocated data points within each region is roughly ~180,000. However, the AIRS and MODIS cloud fields have smaller spatial resolutions that are aggregated to the AIRS/AMSU field of regard, and the raw counts for these fields number in the millions*."**

This comment definitely applies to the joint pdfs, too. Are the color bars different for the different regions, and how much data is in a black region compared to a white region?

**The gray scale/color contours are exactly the same for all regions to facilitate comparison. The gray scale indicates the log(count), where black is log(2), about 8, and white goes to log(8), about 3000.  A gray scale bar at the bottom of every joint pdf figure has been added for clarity.**

2. Using three moments of the reflectance is interesting, but the physical interpretation gets lost in the text.

**Reviewer #2 made similar comments in regards to the connection between "cloud organization" and "skewness".  We have de-emphasized "cloud organization" and replaced with "cloud variability" for several occurrences in the manuscript. We have revised the discussion of skewness in Section 4.1 in the revision.**

It is made clear that skewness increases in cumulus regimes as ECF drops. Is the interpretation that this is a measure of cloud size?

**With regard to the interpretation of reduced ECF because of smaller cloud size, that is a great question but we cannot provide an unambiguous answer (but we detail a**

**response below). There are likely several factors at play.**

**We have added some additional text in the middle of Section 4.1 on p. 7 lines 15-28 to highlight these issues: "*There are several factors that contribute to relationships between ECF and the various moments of radiance. A reduced ECF and increased radiance skewness (Fig. 3) may indicate smaller cloud sizes but is probably not universally true. If the cloud optical thickness is decreased, the ECF is also decreased from reductions in cloud emissivity even though cloud coverage itself may remain constant. (Recall that the ECF is a convolution of emissivity and cloud fraction.) If the cloud optical thickness is fixed, the cloud emissivity remains fixed even though the cloud coverage itself and ECF could be decreased. The ECF could also be decreased (increased) if small cloud elements become more widely spaced (packed together) assuming the cloud sizes of the individual cumulus elements remain the same. With respect to the visible radiances, the radiance is decreased if cloud elements become smaller than the nominal 2.2 km pixel size assuming the optical thickness of the cloud elements do not change. Therefore, if an increased proportion of a cloud population with normally distributed radiances becomes subpixel in size, one would expect a shift towards positive skewness. If cloud distributions are spatially resolved, an increased skewness radiance is still entirely possible if the true optical thickness of cloud distributions is skewed itself. However, in this investigation, the skewness of the MODIS optical thickness is less skewed at low ECF than visible radiance (not shown). This suggests that the skewness in the visible radiance at low ECF is at least partially caused by smaller cloud sizes.***"

The standard deviation of reflectance seems to be connected to the boundary layer depth (Fig 8 & Page 9). Is that expected? The standard deviation isn't used much except to make this point, and it is not clear that it adds much to the overall story. Maybe it would be worth extracting the standard deviation of reflectance into supplemental material?

**Indeed this is the case. We have slightly modified the relevant text on p. 10 lines 7-9 to clarify that the MBL depth's relationship to the standard deviation of radiance is generally linear, while with the mean radiance it is nonlinear (the maximum MBL depth is in the middle of the joint pdf). We agree that the details of standard deviation may be explored more completely in future work. This particular point about the linearity of nonlinearity between the moments is rather interesting and worth describing, albeit briefly. We have decided to leave Figure 8 as is.**

3. Section 4.2 is lacking. If I understand correctly, the point of this section is to whittle down the number of variables to look at in the later sections, settling on reflectance and ECF as the "phase space" (or maybe the "independent" or "predictor" variables?). The weakness is that the selection seems to be mostly arbitrary rather than by systematic evaluation.

**We agree that the starting point for dimensionality choice appears to be pretty arbitrary in the manuscript but our approach was not. We draw upon a response to reviewer #2 to partially address this concern:**

**"The most honest way to go into this investigation is to ask what do we do when confronted with a choice from an enormous selection of available data? Dozens of geophysical variables are available from each instrument and reanalysis system. These variables can be plotted against each other in 1000s of combinations (or much more). The moments of these variables are also another dimensional choice, so to speak. On top of that, any field can be overlaid onto these two dimensions as done in the figures. So where does one start? As we pointed out, our reasoning for starting where we did is found here:**

**Line 3, page 8: "Motivated in large part to link cloud and thermodynamic properties derived from infrared and visible bands…"**

In the list of comparisons, two combinations are missing that would fill in the matrix: visible reflectance and tau; ECF and cloud fraction. Both of these seem like they would exhibit strong correlations, so maybe that is why they are omitted. But in other parts of the text there is a contrast made between the MODIS and AIRS cloud fractions, so seeing ECF versus cloud fraction would be useful.

**We have added these two dimensional choices to a revised version of Fig. 7. We have added them in order to enhance the discussion in the challenges of choosing the appropriate dimensionality. One combination (radiance and tau) exhibits a strong correlation while the other combination (CF versus ECF) exhibits a poor correlation. The following text has been added to the middle of Section 4.2 on page 9 lines 15-19: "*The two other panels highlight the challenges with the choice of dimensionality. In the case of radiance versus $\tau$, while there is a strong correlation in the occurrence frequency in the more reflective clouds, the structure in the MBL depth is much less clear. In the case of cloud fraction versus ECF, the occurrence frequency is much more poorly correlated and scattered, while the MBL depth shows less structure in either dimension.*"**

That accounts for an assessment of the "phase space" variables, but MBL depth is also being discussed here, but it is not clear why. Is the MBL standing in here for an integrated "thermodynamic" variable?

**MBL depth was chosen as a representative variable just to make the point of why the dimensions were chosen. We want to focus on dimensional choices that show structure in fields such as MBL depth. The same plots with RH, dMSE, etc. were also made and the story is very similar. The overlying quantity in the joint pdf has a larger dynamic range when ECF (infrared) and radiance (visible) is used.**

4. The comparison of the regions. Early in the paper (Sections 1-3), it makes sense to look at the four regions separately. Going through Section 4, my feeling is that mostly the NEP, SEP, and SEA act very similarly, while NEA is an outlier. This is likely due to the NEA being more strongly influenced by midlatitude systems (even when filtered for mid and high clouds). A few points are raised about the difference between the hemispheres,

but it isn't clear whether there is enough sampling (especially with only one season) to make any definitive statements. So I wondered, especially at the end of Section 4.3, whether it would simplify things to combine NEP, SEP, and SEA into one population and exclude NEA or show it as a contrasting population? The advantage is to reduce figure panels and increase overall sample size at the expense of having a comparison of the regions. In the present form, I don't see that the bottom line of the paper is really emphasizing any differences in the regions except that NEA is different from the others. As a related note, the title of Section 4.3 is "Regional differences in MBL depth and dMSE," but my main takeaway from Figure 8 is the similarity of the regions, and I felt like dMSE was not much emphasized in the section.

**The sample sizes are now added to Table 1 and the whitish areas of the joint pdfs have counts that number in the 100s to a few 1000s.**

**Thanks for the suggestion of combining the three regions. Some of the differences between the NEP, SEP, and SEA are fairly robust. For instance, the SEP and SEA have larger re, stronger u925, a more clear relationship with omega700 and omega925, and a deeper MBL than the NEP. These observations are consistent with what we currently understand about the regional differences. If the three regions are combined, these patterns will be averaged over and smoothed out. The counts in each region are found in different portions of the ECF and radiance dimensional space and will distort the patterns. (For example, the SEP has many more points at higher ECF than the NEP.) Ultimately, we are concerned that a three region joint pdf would more poorly resemble each of the three regions individually.**

**We have added the following text to p. 10 lines 6-7 to reiterate the reviewer's point about the similarity among three of four regions: "*Generally speaking the NEA is the largest outlier of the four regions for all radiance moments shown for MBL depth in Fig. 8 and is more affected by the midlatitudes than other regions.*"**

**We have revised the title of Section 4.3 to "*Regional similarity in MBL depth*".**

5. Comparing different scales. This study focuses on the smallest scales possible for the data, which is interesting by itself. There should be some care taken when comparing to previous studies that are explicitly working at much larger scales. This comes up in a few places in the text, but prominently at the end of section 4.3 where there is a conclusion that dMSE is correlated with small-scale spatial structure *rather than* large-scale thermodynamic structure. This might be misleading. When averaged up to longer time scales, it seems reasonable that dMSE is more representative of the large scale thermodynamic structure than the spatial structure of clouds. The same holds for LTS and EIS; the relationships between these bulk measures of inversion strength and cloud cover are only valid on relatively long time scales. Recall that the Klein-Hartmann line is derived using seasonal averages. This is discussed occasionally in the literature; one example is found in Zhang et al. (2009, DOI:10.1175/2009JCLI2891.1) where they point out that sampling the low-level divergence distribution is important for capturing the relationship between LTS and cloud cover.

**Thanks for pointing out the time and spatial scale context of the agreement. In the revision, we have changed the text in question on p. 10 lines 32-33 and p. 11 lines 1-2 at the end of Section 4.3 to the following: "*This behavior is similar to MBL depth (Fig. 8f) and suggests that instantaneous values of dMSE correlate well with small-scale cloud variability. This is not inconsistent with LTS and dMSE correlating well with larger-scale atmospheric thermodynamic structure on much longer time scales.*"**

6. Value of Section 4.4? The text seems to suggest that the point of this section is to compare AIRS and MERRA RH, showing they are similar and therefore useful. The MERRA RH isn't shown here (added as Figure A1), which undercuts this as the main message of the section. The section title is just "vertical structure of RH," but it is pretty hard to get a good sense for the vertical structure from the conditionally averaged contour plots showing one level at a time. The question is what aspect of the RH structure is needed to advance the overall argument of the paper? Based on Section 5, it is not clear that the vertical structure of RH is integral to the paper and Section 4.4 and Figure 10 could be deleted.

**We agree with the reviewer that this section is tangential and we have eliminated it in the revision. The reason it was originally included (along with the Appendix) is that it pointed out unambiguously the nature of infrared sounding in and around clouds within the MBL. In the response during the discussions phase, we thought it might be best to combine into the Appendix and keep the figures, but after further consideration, we completely agree with the reviewer and will remove it since it does not advance the investigation.**

**The lines of deleted text with regard to this revision are found here: p. 4 lines 32-33; p. 5 lines 1-2; p. 8 lines 13-14 and 24-26; p. 11 lines 8-15 and 24-25; p. 14 lines 24-25; p. 15 lines 3-8; p. 16 lines 1-9.**

7. The connection to microphysical effects. Section 4.5 brings r_e into the picture, and suggests that the difference between the stratocumulus and cumulus is due to microphysical processes. The next section makes the connection to wind speed, which is interesting. I'm not sure I understand the physical interpretation of the result.

**The physical connection between wind speed and effective radius is drawn out more carefully in the new Section 4.4. In short, the stronger MBL wind is related to a deeper and moister MBL with more frequent precipitating clouds, which is observed as larger re in MODIS data. This could be caused by increased subpixel inhomogeneity or larger re in the cloud.**

Also, it seems like making the link via the comparison of the contour plots in Figures 11 e-h and 13 e-h is a little cumbersome. Does viewing this relationship within the reflectance cloud fraction phase space make the most sense here, and if so, what do we get from this view that would not appear by directly correlating r_e and u925, for example? This seems like a key finding in the paper, and it might be better drawn out by

combining sections 4.5 and 4.6 into a more unified discussion of the r_e variation and connection to meteorology and microphysical processes.

**This is a good idea and the former Sections 4.5 and 4.6 are now unified into a single Section 4.4 titled '*Relating meteorology and microphysical processes'*. Some of the front matter to the old Section 4.6 now starts this new combined section.**

**We contend that the radiance dimension is very useful for the discussion of r_e and meteorological variables and will leave that dimension as is for the revision. A big reason why we wanted radiance for r_e is that it is easier to point out the subpixel inhomogeneity and 3-D radiative transfer issues that arise in the highly skewed portions of the joint pdfs with low ECF that are discussed in section 4.4.**

**We also carried out a few additional calculations to investigate direct correlations between u_925 and reff and below we show the 2d histogram for the SEP region during SON. The binning in reff is 0.5 microns while the binning for u_925 is 0.5 m/s. The minimum count for black=10 and the maximum count for red=450.**

[Figure]

**There is a relationship between u_925 and reff in the absence of other parameters such as ECF and radiance.  The correlation appears to have two regimes: a more strongly sloped one at lower values of u_925 and reff, and a less strongly sloped one at higher values.  We further point out that this plot is not directly comparable to the results of Figs. 10 and 12 for the same quantities because each value that populates a given bin in the above figure can be found throughout the radiance and ECF dimensions. The fact that the figures in the paper appear to show a stronger correspondence in the radiance and ECF dimensions than shown above further suggests the importance of the context of the cloud amount in which this correlation operates.**

Anonymous Referee #2

**We thank the reviewer for taking the time and effort to review the manuscript and appreciate the detailed and thoughtful comments. In this response, we aim to highlight more clearly the purpose, value, and novelty claimed in the manuscript.**

**For this particular final response, we have added specific references to line and page numbers with respect to the track change version, figure and table changes, and additional comments on top of those provided during the discussions phase. In the track change version, all deletions are marked with a  and all additions are marked in red.**

**Note that we have changed 'reflectance' to 'radiance' throughout the manuscript and use the latter in the responses to be consistent with the units reported in the AIRS visible Level 1 files (Watts/meter\*\*2/micron/steradian).**

General comment and recommendation

This manuscript presents a comparison of (correlations between) cloud properties and thermodynamic and dynamic fields derived from AIRS and MERRA data, with those derived in previous literature and derived in this paper from MODIS.

**This paper is not a comparison or a validation paper. Its purpose is to describe the synergistic use of previously validated data products from AIRS, MODIS, and CloudSat, together with MERRA reanalysis at the native temporal and spatial resolution, to investigate relationships between cloud microphysical and optical properties, and dynamical and thermodynamic fields. To our knowledge, at the time of submission, we have not seen this type of approach for MBL processes. We have added the following text on p. 3 line 15-16 to clarify: "*Our primary purpose is to investigate instantaneous relationships between cloud microphysical and optical properties, dynamical, and thermodynamic variables fields from the A-train and MERRA at the native temporal and spatial resolution of the observations*."**

The manuscript comes across as rather unfocused, wandering between a variety of objectives, none of which end up convincingly presented.

**This is a fair statement. We have tightened up the organization of the various components in the manuscript, eliminated the RH results, trimmed some of the introduction and references, and are clearer with regard to the conclusions and take home messages.**

The manuscript appears to: a) evaluate AIRS and MERRA against MODIS and other cited products;

**As stated above, this paper is not a comparison, evaluation, or validation paper. In fact, the only common field used is RH between AIRS and MERRA. Section 4.4 and**

**Appendix A are removed from the revised manuscript per reviewer #1's suggestion and reviewer #2's concern about meandering between objectives.**

**AIRS, MODIS, CloudSat, and MERRA each provide unique information that can be brought to bear on observing the subtropical MBL. We are not comparing common geophysical fields obtained between them. Our guiding philosophy is "Why not play to the strengths of each instrument?" We have added text on p. 3, line 23-25: "*The satellite and reanalysis data each provide unique information that should ideally be combined together at the native resolution rather than relying on one instrument or reanalasys alone.*"**

b) provide physical insights into what explains the transition by means of reflectance, optical depth, boundary layer depth and effective radius;

**That is (partially) correct, although there are many other geophysical variables used, and various moments (mean, variance, skewness) that highlight certain aspects of MBL structure. However, we are not attempting to "explain the transition" so much as rather present a new way to observe it, and more generally all cloud regimes (although others are beyond the scope of the investigation).**

c) present the AIRS cloud product that can best reflect the transition from stratocumulus to cumulus, where a variety of measures are tried out;

**There are three cloud products from AIRS that are used. (1) The AIRS cloud thermodynamic phase product is used to coarsely group together uniform stratocumulus and broken shallow cumulus. (2) The AIRS effective cloud fraction (ECF) is derived from infrared channels so it will have a different perspective of cloud cover compared to visible reflectance or optical thickness. (3) The AIRS visible channels are used to quantify the reflectance that is filtered by an AIRS visible cloud mask.**

d) present different physical behaviours between four regions in which the transition between stratocumulus and cumulus occurs.

**We did not attempt to explain why the four regions exhibit such differences and similarities in the transition itself. This is well beyond the scope of this investigation and would require extensive numerical modeling experiments and further investigation of several years of data. The following text has been added for clarification on p. 8 line 33 and p. 9 lines 1-2: "*While the variability within each region and between the four regions is consistent with previous studies, the physical reasons for these differences are beyond the scope of the current investigation.*"**

Sometimes the lack of focus and a specific question of interest seems to shines through in the authors' writing, for instance, when they introduce new sections, which might represent choices that are "not optimal", but simply provide "a fresh look at available products",

**It appears the reviewer is referring to line 13, page 8 of the submitted version of the manuscript for the quoted text. We offered specific reasons for why we chose the effective cloud fraction (ECF) variable from AIRS over the cloud fraction (CF) derived from the MODIS cloud mask, and also radiance from AIRS over the optical thickness from MODIS. The reasons are described just above the quoted text:**

**lines 7-11, page 8: "*The MBL depth exhibits clearer patterns in the ECF dimension rather than the cloud fraction dimension. The latter is more compressed and the gradients are weaker in both dimensions. The MBL depth is deepest for lower values of ECF, $\tau$, and reflectance. In addition, the MBL depth also decreases for the most reflective clouds at a given value of ECF while this behavior is not observed for $\tau$. We posit that an additional population of sub-pixel cumulus clouds is captured within the reflectance data that is not captured in $\tau$ data.*"**

or when they describe that choosing which variables to plot in their joint pdfs is challenging.

**It appears the reviewer is referring to line 2, page 8 of the submitted version of the manuscript. The most honest way to go into this investigation is to ask what do we do when confronted with a choice from an enormous selection of available data? Dozens of geophysical variables are available from each instrument and reanalysis system. These variables can be plotted against each other in 1000s of combinations (or much more). The moments of these variables are also another dimensional choice, so to speak. On top of that, any field can be overlaid onto the two dimensions as done throughout the joint pdf figures. So where does one start? As we pointed out, our reasoning for starting where we did is found here:**

**Line 3, page 8 (of submitted version): "*Motivated in large part to link cloud and thermodynamic properties derived from infrared and visible bands…*"**

**Reviewer #1 also had comments on this section under their point (3) and we have added two additional panels to figure 8. Please see this response to reviewer #1.**

**We have removed 'a fresh look' and modified text around p. 9 lines 21-22.**

If the focus would be on presenting novel insights,

**Since the reviewer is emphasizing multiple times the "novelty" of this work, after doing a word search we found only three instances of the word "novel" are used in the manuscript. This word choice is unfortunate and we have removed the three occurrences of novelty in the revision.**

I had expected that beyond abstract descriptions of behavior of different quantities in the joint pdf's the authors explain what this behaviour actually tells us about observed cloud fields.

**The revised Section 4.4 'Relating meteorology and microphysics' goes into detail about the behavior between multiple independent cloud variables and other properties. We show that the MODIS reff corresponds closely to u_925 and moistening and deepening of the MBL. This study is consistent with surface observations taken during the RICO campaign.**

**This paper is also about an attempt to describe a more holistic synthesis of the subtropical MBL from the point of view of A-train satellite observations and MERRA reanalysis built from native temporal and spatial resolution data. This paper is not about the physical causes of the stratocumulus to cumulus transition, but we do cite some of these papers in the Introduction as motivation.**

If the focus would be on an evaluation of AIRS products, I would expect that the evaluation were more thorough and go beyond a comparison of seasonal averaged fields.

**As stated above, this paper is not an evaluation paper for AIRS products. We have cited references throughout that point the reader to previous validation efforts that support the use of the data as shown in the manuscript.**

What contributes to the wandering is that the authors use different data sets for different objectives as they present themselves.

**The whole purpose of the paper is to use the different instruments and reanalysis data sets as building blocks to construct a simultaneous point of view of the MBL, playing on the strengths of each instrument. The reviewer comment strongly suggests we need to be much more clear and concise about our purpose. We hope that this concern has been resolved with the revisions.**

AIRS and MERRA are used for interpreting skewness measures and the transition between cloud types, along with comparisons of MODIS. Only MODIS is used for the purpose of evaluating effective radii in the two regimes, and what might physically or methodologically explain effective radii behaviour.

**As far as we are aware, MODIS has the most useful, validated, tested, and investigated global retrieval of liquid water cloud effective radius available to the scientific community. AIRS does not provide one. CloudSat uses MODIS effective radius in its forward algorithm of retrieval products. MERRA is quite awful at clouds.**

In much of the authors assertions, previous literature is referenced, but often not explained.

**We tried to be as comprehensive as possible with citing references for our statements. In the revision we have tried to be as clear as possible as to why we are**

**citing a particular work, and also eight references have been removed.**

**The lines of deleted text with regard to this revision are found here: p. 2 lines 17-21; p. 3 lines 5-6 and 13-14; p. 4 lines 3-5; p. 5 lines 22-23.**

**The deleted references are: Atkinson et al., Brient et al., Christensen et al., Nasiri et al., Platnick et al., 2003 (we keep 2017), Sandu and Stevens, Tian et al., and Vergados et al.**

Because the manuscript does not present a novel insight and fails to convincingly argue for any method or the AIRS or MERRA datasets at providing novel insights, I recommend a rejection of the manuscript.

**We would like to bring up that reviewer #1 had a different opinion: "This effort is commendable and valuable, combining these data at small scales can potentially reveal a lot about the relationships between clouds and their environment."**

**We hope that we will convince reviewer #2 the value of this work with the revisions that are made.**

Specific comments

The title is unspecific (which satellite and reanalysis data sets?) and promises more insight than the paper offers. The term cloud organization is not well-chosen, because the authors do not present results on cloud organization nor discuss what cloud organization means. The words organised and disorganised are repeatedly used throughout the manuscript, but mostly in reference to organised stratocumulus and disorganised trade-wind cumulus. Both stratocumulus and trade-cumulus can be organized and disorganized, depending on some definition of organization. Sometimes it seems the authors refer to homogeneous and heterogeneous, but mostly it seems that they refer to the two different cloud types.

**We have changed the title to "An A-train and MERRA view of cloud, thermodynamic, and dynamic variability within the subtropical marine boundary layer". Thus the satellite and reanalysis data are more specific and we have removed the word 'organization'.**

**The reviewer is entirely correct with respect to organization. That was an imprecise way to describe the data. The same exact values of mean, standard deviation, and skewness of cloud fraction, reflectance, and other fields can exhibit organization in a multitude of manners. Therefore, we will emphasize the characteristics of the moments rather than the organization. We have responded to one of reviewer #1's concerns about the use of moments under their point (2). We have added discussion regarding the interpretation of positively skewed reflectances. Please refer to this response.**

The word novel is repeatedly used, but seems an overstatement. In much of the manuscript, the authors confirm insights found in previous studies, and that citation list is long.

**We found the word 'novel' was used only three times in the entire manuscript and have reworded to deemphasize that the results are novel.**

**We argue that it is a strength of the approach taken that a long list of previous findings are reaffirmed. Given how many data sets are involved and the extensive nature of research on this topic, we think that the citation list is appropriate. Eight of the references have been removed in response to concerns from both reviewers.**

One novelty that is argued for is the use of AIRS and MERRA datasets at their instantaneous native resolution. But to prove the suitability for these datasets for this kind of study, the authors qualitatively compare the morphology of the stratocumulus to cumulus transition from seasonal averaged AIRS and MERRA data with the morphology known from existing studies. I do not think a qualitative comparison of the seasonal mean transition tells us enough about how good AIRS and MERRA perform at their native resolution.

**The seasonal averages were developed as a first order check on our methods from the pixel-scale instantaneous matches from which the seasonal maps were generated.  The seasonal averages are one of the only ways to compare with previous studies given that the previous studies typically show seasonal maps of MBL properties.**

**The individual cloud and thermodynamic properties from AIRS and MODIS have been previously validated and evaluated at the pixel scale and are described in the cited references.**

The authors make an argument for separating the cloud regimes stratocumulus and cumulus based on infrared-based thermodynamic phase (rather than by dynamical regime such as done in previous literature).

**The motivation for this approach is found in the Methodology section. As this is a pixel based approach, we require that all ice cloud instances are removed, and we are confident that the AIRS phase product is more than sufficient and supported by the cited references about its validation and previous uses.**

**Page 5, lines 15-16: "*Removal of pixels containing mid- and high-level clouds helps to reduce ambiguities introduced by free tropospheric clouds and also a portion of the thermodynamic and dynamic variability associated with cloudy areas of synoptic-scale waves.*"**

**The dynamical approach is consistent with this approach in the sense that stratocumulus clouds show larger free tropospheric subsidence than the cumulus**

**clouds. Below we have added a figure for the response.**

[Figure]

*Figure caption: SEP region visible radiance versus effective cloud fraction for cumulus scenes (lower row) and stratocumulus scenes (upper row) at 700 hPa (left column), 850 hPa (center column), and 925 hPa (right column). Omega is overlaid as colored contours and is in units of hPa/day.*

**The omega925, omega850, and omega700 are all larger in the case of stratocumulus than cumulus. In fact, there is a gradient in omega at all three levels for cumulus in the reflectance dimension that is not observed in stratocumulus. By separating the two cloud types this gradient is observed. More reflective clouds appear to be associated with weaker subsidence in the cumulus regime.**

The thermodynamic phase provides information about whether just liquid or ice is present in the detected clouds.

Based on a single scene in Figure 1 and 2 the authors argue that stratocumulus is well identified by those pixels that are detected as liquid, whereas trade-wind cumulus are those pixels that have an unknown thermodynamic phase. How do the authors know that this separation holds well for other scenes?

**The granule maps in Figures 1 and 2 are shown to illustrate the data and methods. We have explored the characteristics of AIRS thermodynamic phase for the entire AIRS record and have published several papers on the topic. AIRS is a radiometrically very stable instrument with very strong sensitivity to cloud phase as discussed in Kahn et al. (2014) and Jin and Nasiri (2014) and citations within.**

**We refer the reviewer to page 5, lines 20-22: "*Jin and Nasiri (2014) showed that AIRS successfully identifies the presence of ice within the AIRS FOV in excess of 90% of the time when compared to CALIPSO thermodynamic phase estimates.*"**

After all, trade-wind cumulus are also made of liquid only, and it is unclear and not explained why they could not be identified as such in other scenes. It is also not clear for what purpose the two cloud types are separated here in this paper. Mostly this seems to be a proposition to use AIRS thermodynamic phase in future studies, but with insufficient evidence.

**We agree with the reviewer that the use of the liquid and unknown phase categories is a confusing aspect of how the data is used. The delineation between stratocumulus (liquid) and cumulus (unknown) is made clearer in the revision. The following text has been added to p. 5, line 26-31: "*As the AIRS cloud phase algorithm is based on a channel selection that exploits differences in the index of refraction for liquid and ice, it is possible that the cloud amount observed in the AIRS pixel is small enough the spectral signature is small such that it does not trigger a positive liquid test (e.g., Jin and Nasiri, 2014). The ECF for these unknown phase cases can be above the sensitivity of cloud detection (validated using CALIPSO lidar, see Kahn et al., 2014). As a result, none of the phase tests are triggered even though a small amount of cloud is found in the AIRS pixel. These unknown cases line up very well with the frequency of trade cumulus in the four regions selected.*"**

One aspect of the paper that prevents it from providing clear physical insights (if this were the main objective) is that the authors never explain what the skewness in reflectance or optical depth tells us about the nature of the cloud field that is observed (and this is true for many of the behaviours derived from the joint pdfs). The skewness measure has been used in previous studies, and can with some background of course be interpreted, but the authors never explicitly do. This makes the description of results rather abstract.

**We agree with the reviewer that this is a weakness in the paper. Thus, we have strengthened this aspect of the interpretation of the data. On the whole, the more skewed the reflectance is, the smaller the ECF is. When the reflectance is approximately Gaussian, the ECF is larger. The former is seen very clearly in the cumulus pdfs and the latter in the stratocumulus pdfs. Since there is such good separation between the two cloud types, they should be discussed separately. This also should be considered as an independent confirmation of the sensitivity of the AIRS phase algorithm to cloud type.**

**Even for the same combination of reflectance and ECF in cumulus and stratocumulus pdfs for the MBL depth, the MBL depth is shallower for stratocumulus. The same is true for dMSE (more positive for stratocumulus than trade cumulus.) This is a really interesting result that shows there is cloud regime dependence even for the same value of ECF and reflectance, and that separation is**

**facilitated by the AIRS phase algorithm categories liquid and unknown.**

**The text has been revised accordingly. As reviewer #1 had similar concerns, please refer to the response to reviewer #1 under point (2) for additions and changes made to the manuscript.**

The discussion in section 4.5 and the conclusions argue for both physical causes (precipitation) as well as retrieval-related biases (inhomogeneity) for the observed larger effective radius in cumulus clouds compared to stratocumulus. But whereas first is stated that (L24) "the observed increase in re is entirely consistent with environmental variability (winds/droplet growth/precipitation)", it is written further along that the greater inhomogeneity in such precipitating cumulus fields can cause assumptions used in retrievals to break down. Hence, should I trust the retrieved larger effective radii observed?

**The work of Cho et al (2015) goes into great detail on the failures in the MODIS re retrievals and the various causes. The factors that cause the failed retrievals, discussed in Cho et al., are also operating within the successful retrievals described in this paper. The bottom line is that the pixel-scale inhomogeneity is producing larger values of re for successful retrievals that are correlated with increased MBL wind and precipitation frequency. Cloud inhomogeneity is correlated to precipitation and that is a big reason why the re increases. However, the actual cloud droplets can be larger too. The underlying assumption about the cloud droplet size distribution may also be problematic and other factors may come into play. Teasing apart these effects warrants significant research efforts; these questions are being pursued by the MODIS algorithm team.**

**We have made several modifications to the new Section 4.4 that includes discussion of re and its interpretation:**

**p. 11, lines 30-31: "*Figure 10 shows the MODIS derived $r_e$ for stratocumulus (Fig. 10a-d) and cumulus (Fig. 10e-p) that are limited to successful retrievals (no PCL pixels are included).*"**

**p. 12, lines 3-4: "*Cloud inhomogeneity may also lead to significant 3-D radiative transfer effects but these tend to cause both larger and smaller $r_e$ in similar proportions (Zhang et al., 2012).*"**

**p. 12, lines 11-15: "*One general interpretation of the larger $r_e$ in cumulus (Fig. 10e-h) when contrasted to stratocumulus (Fig. 10a-d) is that it is caused by increased inhomogeneity of cumulus (Zhang et al., 2012), retrieval failures and partly cloudy pixels (Cho et al., 2015), and view angle biases (e.g., Liang et al., 2015) that are further coupled together with other factors at play (Zhang et al., 2016). The aforementioned issues may still impact a successful $r_e$ retrieval.*"**

**p. 12, lines 20-22: "*Successful retrievals with pixels that have increased subpixel***

*horizontal inhomogeneity may be more frequently precipitating, either because of larger $r_e$ in the cloud, or because the plane parallel homogeneous bias is larger in precipitating clouds.***"**

**p. 15, lines 13-15: "***This may be caused by larger $r_e$ in the cloud itself or that precipitating clouds are associated with an increased subpixel inhomogeneity that leads to the plane parallel homogeneous bias; this topic warrants further investigation.***"**

**The lines of deleted text with regard to this revision are found here: p. 4 lines 21-23; p. 15 lines 15-19.**

In the last paragraphs of section 4.6 and the summary, the authors argue two seemingly contradicting statements with which they end their manuscript. Namely, that three of the four regions studied show similar relationships and behaviours among cloud-related quantities and the (thermo)dynamic state, but also that the relationships are non-unique (can vary greatly), for which their datasets provide a good opportunity for further exploration. I understand the subtlety, but is this the best ending?

**We agree that this isn't the most useful of endings. We have revised in the end of the new Section 4.4 (which was previously Section 4.6) and removed the sentence in question. In the summary, we have removed the last two sentences. Now the summary ends with the following short paragraph on future work on p. 15, lines 28-31: "***
[revised manuscript text omitted]

[Figure]

**Figure 12 . Joint pdfs of 700 hPa θe, u925, ω700, and ω925,  for the four regions listed in Table 1.**

---

## Author Response (AR2)

**The authors again greatly appreciate the constructive comments of the two anonymous reviewers. Our responses and modifications to the manuscript are shown below in bold text.**

Referee #1

The revised manuscript is much improved. There is much greater focus on presenting these observations and highlighting the potential for this kind of investigation at the smallest possible scales from satellites and reanalysis. Besides a few minor comments that could be addressed and some typographical errors, this paper appears suitable for publication. As was clear in the original version, the great strength of the paper is the excellent collation of observations and reanalysis to provide an interesting view of subtropical clouds. The findings appear generally in line with previous results, but this paper provides a good overview, and emphasizes smaller scales than most other studies. Probably the biggest weaknesses in the paper are: (1) the introduction does not connect very well to the rest of the study, and (2) the emphasis on the joint histograms (Figs 7-12) might be too strong. These are subjective, and maybe my take on these is different from another reader.

**We have made modifications in response to both of the stated weaknesses. Please refer to below responses.**

Minor Comments

1. In terms of the introduction, I think I understand what the text is trying to do. There is a lot of discussion and reference to cloud feedbacks and climate modeling, and the point might be that we need to better understand basic processes in nature to improve our understanding and prediction of the future. I found that by the end of the introduction, however, I did not have a very good understanding of where this paper was really heading and why it matters. That is to say, why it matters in relation to similar literature rather than the big picture of cloud feedbacks and climate change prediction. This is a minor comment, so the authors should not be compelled to change it, but my feeling is that this paper would be better served by condensing the present introduction to a paragraph or two and use the recovered space to provide more context for this study.

**We revisited each paragraph (and sentence within each paragraph) and feel as though the first five paragraphs of the introduction are appropriate and relevant, but the sixth needs to be improved. The first paragraph briefly calls out the cloud feedback problem with respect to MBL clouds. The second paragraph briefly hits on the physical mechanisms of the MBL feedback problem in the subtropical MBL and key references, with particular attention to the work of L. Nuijens, which we follow up with in the results section. The third paragraph then touches on the vertical structure of thermodynamics (e.g., RH) and how further examination of observations is warranted. The fourth paragraph links together the vertical thermodynamics, cloud structures and variability, and cloud statistical moments, which will be quantitatively addressed in this work. The fifth paragraph follows on**

**with cloud regime dependence of these behaviors (e.g., cumulus versus stratocumulus) and how A-train and reanalysis data can be further exploited.**

The last two paragraphs of the section start to do that, but don't quite get there. For example, in the last paragraph (neglecting the "outline paragraph"), the purpose of the paper is stated in the first sentence, but that is followed by a summary of the methodology (page 3, lines 5-9) which could be deleted because they will be stated in more detail in the next section. A later sentence ("The geophysical fields ...") is also a little too much like methodology, but it could be rewritten to provide more information about why it is important to the goal of the paper to investigate these scales and to examine them as distributions.

**We agree that the second to last paragraph can benefit from significant modification. As suggested, we moved lines 5-9 out of the paragraph to the end of the Methodology section after description of each data set. We then added several lines of text that borrow heavily from the responses to reviews of the previous version of the manuscript as suggested by reviewer #2. The new paragraph at the end of the Introduction now reads as follows:**

*"Our primary purpose is to investigate instantaneous relationships between cloud microphysical and optical properties, dynamical, and thermodynamic variables from the A-train and MERRA at the native temporal and spatial resolution of the observations. The satellite and reanalysis data each provide unique information that should ideally be combined together at the native resolution rather than relying on one instrument or reanalysis alone, or combining over time and space averages. The geophysical fields are retained at the native spatial and temporal resolution such that the instantaneous spatial "snapshots" of the cloud probability density function (pdf) are preserved and are then conditioned by available thermodynamic and dynamic variables. This approach removes the temporal variability in order to focus on the spatial variability and covariances. The statistical behavior of cloud properties, and how the thermodynamic and dynamic state variables are related to them, are thus inferred using the finest temporal and spatial resolutions available. The different instruments and reanalysis data sets are treated as "building blocks" that construct a simultaneous view of the MBL, playing on the strengths of each data set. This holistic synthesis of multivariate and multi-moment data sets may highlight aspects of MBL structure that are otherwise overlooked. The MBL structures of interest are summarized in Nuijens et al. (2009) using surface-based observations and demonstrate testable relationships between clouds, wind, humidity, and precipitation. Lastly, the approach taken herein may ultimately enhance our ability to quantify the complex time, space, and cloud regime coupling of clouds and circulation (Bony et al., 2015)."*

2. It also seems that there is an overemphasis on the joint histograms. While these are nice figures with a lot of information, I often wondered whether they were being used effectively. In particular, the text asks the reader often to compare panels visually. Actually, this is a bit difficult because the data distribution, the gray shading, is the most striking part of the figures while the comparisons are almost always regarding the colored contour lines. It's hard to provide a good suggestion for how to improve this. One possibility is to support the joint distributions with the marginal (i.e., 1-d) distributions; this could be useful especially when the dependence seems less bivariate, like in Figure 1a, 1c, and 1d where the dependence on cloud fraction seems mostly negligible. Another aspect of this over emphasis on the joint histograms is that correlations are left as visual comparisons and are not actually quantified. One could imagine that some of these relationships are not statistically significant because a lot of the variation in the CF-radiance space is occurring far from the core of the data distribution. Similarly, the visual correlations of contour lines across panels could be very misleading (though I don't see any obvious examples where that would be the case); or more importantly, the connections might be made more strongly by making the link visually in the histograms and then bringing the comparisons together in a simpler summary figure that could collect the regions and/or variables into one set of axes. Again, this is a minor comment, but it would be worth considering, especially if the microphysical relationships of Section 4.4 could be more easily summarized in a simpler graphic.

**As done with the Introduction, we went through every subpanel of every figure and feel as though this is the story we want to tell. Each panel is called out in the text and described in brief or at length and help make the points made in the text. The only possible case that might be made is getting rid of the standard deviations in Figure 10 but we did not do so. We intended to highlight differences with the same moment in Figure 8 and we have added some text to highlight the important differences in standard deviation radiance shown between r_eff and MBL depth in Figure 8:**

**"*Fifth, the variations of $r_e$ with the standard deviation of radiance (Fig. 10i-l) are more nonlinear than in the case of MBL depth (Fig. 8i-l). This shows that the relationship with radiance moments is not the same for different physical quantities.*"**

**With regard to marginal 1-d distributions, the general problem is that the information is lost if the data is collapsed in that manner. It also could be another investigation on its own. Since this paper is in large part methodological, we felt it is most germane to not limit the presentation of the joint pdfs and emphasize their multi-moment differences and similarities among the four regions. As highlighted in the text, the differences between regions are large enough such that all four should be shown. The Sc and Cu are different enough that the average radiance should both be highlighted. The standard deviation and skewness reveal more complex behavior, and are thus shown for MBL and r_eff for all four regions.**

3. Page 1, Line 18: "700 an 850" is supposed to be "700-850"

**fixed to 700 and 850**

4. Page 5, Line 3-4: This statement that the unknown cases look like trade cumulus is interesting, but it is not clear whether this is a new result or is included in one of the previous references.

**We have added the following to the end of the sentence to clarify: "…*based on inspection of individual granule data (Figs. 1 and 2) and gridded seasonal averages (Section 4)*."**

5. Page 6, Line 9: This must be NEA instead of NEP

**Fixed**

6. Page 7, lines 10-15: Here and elsewhere, I am not sure dMSE is adding much to this paper as its physical significance does not seem to be explained. Is it just another measure of inversion strength, like LTS and EIS?

**To some degree, yes, it does mimic LTS and EIS, with the effects of vertical moisture gradients included. This is a standard measure used to characterize the MBL and we include it to link to other studies, check our methods, and ultimately show that dMSE can be different between cumulus and stratocumulus for the same ECF and reflectance.**

7. Section 4.1: The tone of this section seems to be more validation than "results." It seems to be mainly showing that expected features are captured. I'm not sure whether that is the intent of the section; if so, then it might be retitled, and if not, then maybe should be revised to highlight new results.

**We changed section 4.1 to its own section 4 and it is titled '*Regional spatial averages*'. Section 5 is now called '*Multivariate and multi-moment pdfs*'**

Referee #2

I thank the authors for their extensive response to my critical review and their revised version. I very much appreciate that they have responded at length at my concerns and made appropriate changes to the manuscript. Their response has been much more convincing of the merit of this manuscript, in particular by more clearly stating the objective and the (limits to the) scope of the paper. I believe the title is a good choice, and I also appreciate the addition of more physical interpretation to what is shown in the plots, for instance, what it means that trade-cu are identified as having an unknown thermodynamic phase.

**We appreciate the efforts of the reviewer to highlight the weaknesses of the manuscript and feel as though the comments greatly contributed to a much clearer manuscript. Many thanks. Peer review works.**

Actually, some of the statements made in the response to reviewers may also be added to the actual manuscript, which would make some aspects a bit more explicit.

**We took the reviewer's suggestion, especially in regard to reviewer #1's comment**

**about strengthening the end of the Introduction.**

Before final acceptance I would ask to consider these:

1. The authors state in their response that they do not attempt to "explain the transition" so much as rather present a new way to observe it. This can be emphasized in the paper itself, and I would even say this more explicitly in the abstract. Rather than "four subtropical transitions regions are investigated", the authors could write something like " we demonstrate a new way of observing the transition - a region of interest because of climate uncertainties - whereby we define stratocu and trade cu exclusively based on infrared thermodynamic phase" etc.

**We have made the following modifications to the abstract.  Those two sentences now read as one: "*A new method for observing four subtropical oceanic regions that capture transitions from stratocumulus to trade cumulus is demonstrated, where stratocumulus and cumulus regimes are based exclusively from infrared-based thermodynamic phase.*"**

2. The authors write in their response that it is challenging to choose which figures to present when confronted with an enormous selection of available data. I understand, but I also think it is the authors task to convincingly show a figure they believe is interesting. I also think that this choice is generally motivated by a specific question: what would you like to answer, and hence, which variables are best to start looking at? The authors may present which questions they are interested in, which in some way they already do by looking at the transition, and in specific by going into more detail about the observed relationship between wind speed, relative humidity and precipitation (effective radius).

**This was definitely the intent of the original submission by ending the description of the results with how similar they were to surface observations shown in Nuijens et al. (2009). We hope that the revised Figure 7 and discussion helps clarify, and the new paragraph in the Introduction makes clearer the intent and outcome of the study.**

Also in the abstract the authors could rephrase their last sentences into something like: "New ways of observing the transition, using the combined Atrain & MERRA dataset, already demonstrate a relationship between effective radius, wind speed and cloudsat precipitation estimates that was previously demonstrated in surface-based observations. Hence, the combined data sets have the potential of adding global context to process-level understanding."

**We have made the following modifications to the abstract.  The end of it now reads: "*The method using the combined A-train and MERRA dataset has demonstrated that an increase in $r_e$ within shallow cumulus is strongly related to higher MBL wind speeds that further correspond to increased precipitation occurrence according to CloudSat, previously demonstrated with surface observations. Hence, the combined data sets have the potential of adding global context to process-level understanding of***

*the MBL.***"**

3. Within the text, the authors could also emphasize that the individual cloud and thermodynamic properties from AIRS and MODIS have been previously validated and evaluated at the pixel scale, and more explicitly say that the seasonal averages are one of the only ways to compare with previous studies given that the previous studies typically only show seasonal maps.

**Further clarification is added in Section 3 near the Jin and Nasiri (2014) reference that the comparisons of AIRS and CALIPSO cloud phase were made at the pixel scale. Further clarification is added in Section 2 near Yue et al. (2013) to point out that AIRS soundings are compared against NWP models and radiosondes at the profile scale. Further clarification is added at the beginning of Section 4 to compare to previous studies.**

[revised manuscript text omitted]